# Mitogenomic evolutionary rates in bilateria are influenced by parasitic lifestyle and locomotory capacity

Ivan Jakovlić [1], Hong Zou[2], Tong Ye[1], Hong Zhang[1], Xiang Liu[1], Chuan-Yu Xiang[1], Gui-Tang Wang[2] & Dong Zhang [1,3] ✉

The evidence that parasitic animals exhibit elevated mitogenomic evolutionary rates is inconsistent and limited to Arthropoda. Similarly, the evidence that mitogenomic evolution is faster in species with low locomotory capacity is limited to a handful of animal lineages. We hypothesised that these two variables are associated and that locomotory capacity is a major underlying factor driving the elevated rates in parasites. Here, we study the evolutionary rates of mitogenomes of 10,906 bilaterian species classified according to their locomotory capacity and parasitic/free-living life history. In Bilateria, evolutionary rates were by far the highest in endoparasites, much lower in ectoparasites with reduced locomotory capacity and free-living lineages with low locomotory capacity, followed by parasitoids, ectoparasites with high locomotory capacity, and finally micropredatory and free-living lineages. The life history categorisation (parasitism) explained ≈45%, locomotory capacity categorisation explained ≈39%, and together they explained ≈56% of the total variability in evolutionary rates of mitochondrial protein-coding genes in Bilateria. Our findings suggest that these two variables play major roles in calibrating the mitogenomic molecular clock in bilaterian animals.

Rates of molecular mitogenomic (mitochondrial genome) evolution are remarkably variable among animal lineages. Previous studies have identified several factors that may be associated with elevated mitogenomic evolutionary rates, but their effects appear to be small, inconsistent, and lineage-specific, which makes mitogenomic evolution difficult to predict[1–5]. Several studies found evidence of elevated mitochondrial evolutionary rates in parasitic lineages[6–10]. However, the underlying cause for this phenomenon remains debated. Previously proposed explanations include (1) directional selection driven by the genetic arms race between hosts and parasites, involving adaptations and counter-adaptations in host-parasite co-evolution[6,11,12], (2) the compensation-draft feedback, where the fixation of mildly deleterious mutations results in the selection for compensatory mutations, which

lead to the fixation of additional deleterious mutations in non-recombining mitochondrial genomes[11], and (3) increased drift associated with reductions in the effective population size ($N_e$)[13], putatively caused by high speciation rates in parasites and/or frequent founder events during transmissions to new host individuals[6,14,15].

Aside from the debate surrounding the factors underlying the association between parasitism and evolutionary rates, the existing evidence for this link is largely limited to Arthropoda[6–10], despite the fact that parasitic lineages have been identified in 15 animal phyla[16]. Among the remaining phyla, there are indications that parasitic and/or commensal Clitellata (Annelida) may exhibit elevated evolutionary rates;[17] however, the dataset supporting this finding was small, and no statistical testing was conducted. In a meta-analysis of all Bilateria

[1]State Key Laboratory of Herbage Improvement and Grassland Agro-ecosystems, and College of Ecology, Lanzhou University, Lanzhou 730000, China. [2]Key Laboratory of Aquaculture Disease Control, Ministry of Agriculture, and State Key Laboratory of Freshwater Ecology and Biotechnology, Institute of Hydrobiology, Chinese Academy of Sciences, Wuhan 430072, China. [3]Key Laboratory of Biodiversity and Environment on the Qinghai-Tibetan Plateau, Ministry of Education, School of Ecology and Environment, Tibet University, 850000 Lhasa, China. ✉e-mail: dongzhang0725@gmail.com

mitogenomes available in 2012, Bernt et al. [18] found exceptionally long branches in several lineages, some of which comprised both free-living and parasitic species (Nematoda, Platyhelminthes, Acari, and Copepoda), but other comprised only free-living (Tunicata and Mollusca). They concluded that parasitism exhibited a somewhat patchy relationship with increased branch lengths[18], but they did not test this statistically either. Over the last seven years, our research team has sequenced over 30 mitogenomes of parasitic animals, primarily focusing on Platyhelminthes and Nematoda, but also including some Arthropoda species (Supplementary Table 1). We have also found some indications that parasites may be disproportionately likely to have rapidly evolving and architecturally destabilised mitogenomes[19-21], but we did not test this statistically either. Furthermore, the association between parasitism and evolutionary rate is inconsistent in arthropod insects: parasitism was associated with an increased evolutionary rate in Hymenoptera, but not in Diptera[15]. Therefore, the association between elevated mitochondrial sequence evolution rates and parasitism is inconsistent (lineage-specific), and it remains statistically untested for taxa other than certain arthropod lineages.

Following the evidence that purifying selection pressure strength is correlated with the selection for locomotory capacity (LOC) in crustaceans[10], we hypothesise that elevated evolutionary rates in some, but not all, parasitic lineages may be driven by a strongly reduced locomotory capacity. The underlying rationale lies in the crucial role of mitochondrial genomes in the production of energy. We posit that the strength of purifying selection acting on mitochondrial genomes should be positively correlated with energy expenditure, which, in turn, should be positively correlated with the strength of selection for locomotory capacity[10]. In endoparasites and some ectoparasites, the near absence of locomotory capacity may allow for a certain degree of degradation in the efficiency of mitogenomic energy production. The existence of a positive correlation between the metabolic rate and purifying selection pressures was observed in salamanders[22], whereas the association between the locomotory capacity and the strength of purifying selection on the mitochondrial genome was confirmed in molluscs[23], crustaceans[10], and in selected insect[24], teleost fish[25,26], bird and mammal[27] lineages. However, the association between the locomotory capacity and evolutionary rate was never comprehensively tested on a dataset comprising multiple major lineages. Furthermore, the impact of parasitism on sequence evolution was never associated with the locomotory capacity to our knowledge.

We therefore hypothesised that the selection of locomotory capacity may help explain the patchy nature of the association between parasitism and elevated mitogenomic evolutionary rates. While the role of $N_e$ in this inconsistency has been recognised previously[15], locomotory capacity appears to have been overlooked by previous studies. Herein we adopted the definition of parasitism as a consumer interaction in which the consumer feeds on a single individual (the host) during at least one life-history stage, where both parasites and hosts belong to the Animalia, which includes strategies employed by parasitoids, parasitic castrators, macroparasites and pathogens, but excludes micropredators, brood 'parasites', kleptoparasites, symbiotic egg predators, inquilines, non-feeding symbionts, and plant-parasites[16,28]. The selection for locomotory capacity is not uniform across different parasitic strategies; whereas endoparasites possess a rudimentary locomotory capacity, certain ectoparasitic (such as fleas) and many parasitoid lineages are evolving under high selection pressures for locomotory capacity. Specifically, parasitoid organisms typically undergo a parasitic larval stage followed by a free-living adult stage, which implies that any degradation in the efficiency of mitogenomic energy production would negatively affect the fitness of the free-living adults. Consequently, their mitogenomes should evolve under stringent purifying selection pressures. Additionally, due to the existence of free-living adults, there is no reason to expect a reduced $N_e$ in comparison to fully free-living species in parasitoids. In contrast, most macroparasitic (comprising endoparasites and ectoparasites) species do not undergo strong selection for locomotory capacity during any of their life stages. Moreover, the evolutionary transition to parasitism often involves a reduction in genomic and metabolic complexity[29-31]. Metabolic dependence on the host may allow a further relaxation of purifying selection pressures[31]. Combined, the loss of locomotory capacity and metabolic reliance on the host may allow strongly reduced metabolic rates in parasites, and consequently relaxation of purifying selection pressures on mitochondrial genes.

Following this reasoning, we anticipate that endoparasites should exhibit the highest evolutionary rates, followed by ectoparasitic lineages with low locomotory capacity. However, we do not expect parasitoids to differ significantly from the free-living lineages. As several previous studies were referring to parasitoids when identifying elevated mitogenomic evolutionary rates in parasites[6,11,15], this might explain the inconsistent results[15] and suggest that elevated evolutionary rates may have been mistakenly attributed to the nominal 'parasitism', when in fact they were driven by other variables.

To test these hypotheses, we conducted a comprehensive study utilizing all available Bilateria mitogenomes from the curated and non-redundant RefSeq database[32]. The dataset comprised around 11,000 species (last accessed on 10th March 2022). We categorised parasitic life histories into endoparasites (comprising mesoparasites; EndoP), ectoparasites (EctoP), and parasitoids. To further test our predictions, we separately categorised micropredators, which are classified as parasites in some categorisations[33]. They resemble many ectoparasites in their blood-feeding habits, but they only interact with the host for feeding, and otherwise they are free-living (examples are mosquitoes and vampire bats). We hypothesised that their evolutionary rates should not differ from other free-living organisms. We also roughly divided the entire dataset into three categories according to the locomotory capacity: Low, Intermediate, and High. To summarise, herein we tested the following hypotheses: 1. Endoparasitic lineages, characterised by the almost complete loss of locomotory capacity, metabolic dependence on the host, and physical confinement to the host, should exhibit the fastest evolutionary rates; 2. Ectoparasites should produce a variable, lineage-specific, signal, since certain ectoparasitic lineages (such as fleas) share similarities with free-living and parasitoid lineages in locomotory capacity and levels of metabolic dependence and physical confinement in relation to the host; whereas others (such as monogenean flatworms) are similar to endoparasites in these aspects; 3. Evolutionary rates should not significantly differ between free-living, parasitoid and micropredatory lineages; and 4. Lineages with low locomotory capacity should exhibit increased evolutionary rates compared to those with high locomotory capacity, regardless of their life history. Furthermore, we compared the relative influence of parasitism and locomotory capacity on the mitogenomic evolution using the complete bilaterian mitogenomic dataset, along with several subsets designed to control for various confounding factors. Our analyses strongly support these hypotheses and lead to the conclusion that both parasitism and locomotory capacity play significant roles in calibrating the mitogenomic molecular clock in bilaterian animals.

## Results

### Pairwise comparisons at different taxonomic levels

The final dataset comprised 10,906 species classified into 25 phyla, a majority of which were Chordata (6228) and Arthropoda (3504) (Supplementary Data 1: Worksheet 1). A vast majority were classified as free-living, but the numbers of parasitoid, micropredatory, ectoparasitic, and endoparasitic species (66 to 276) ensured good statistical power (Cohen's d > 0.96) in all pairwise comparisons apart from

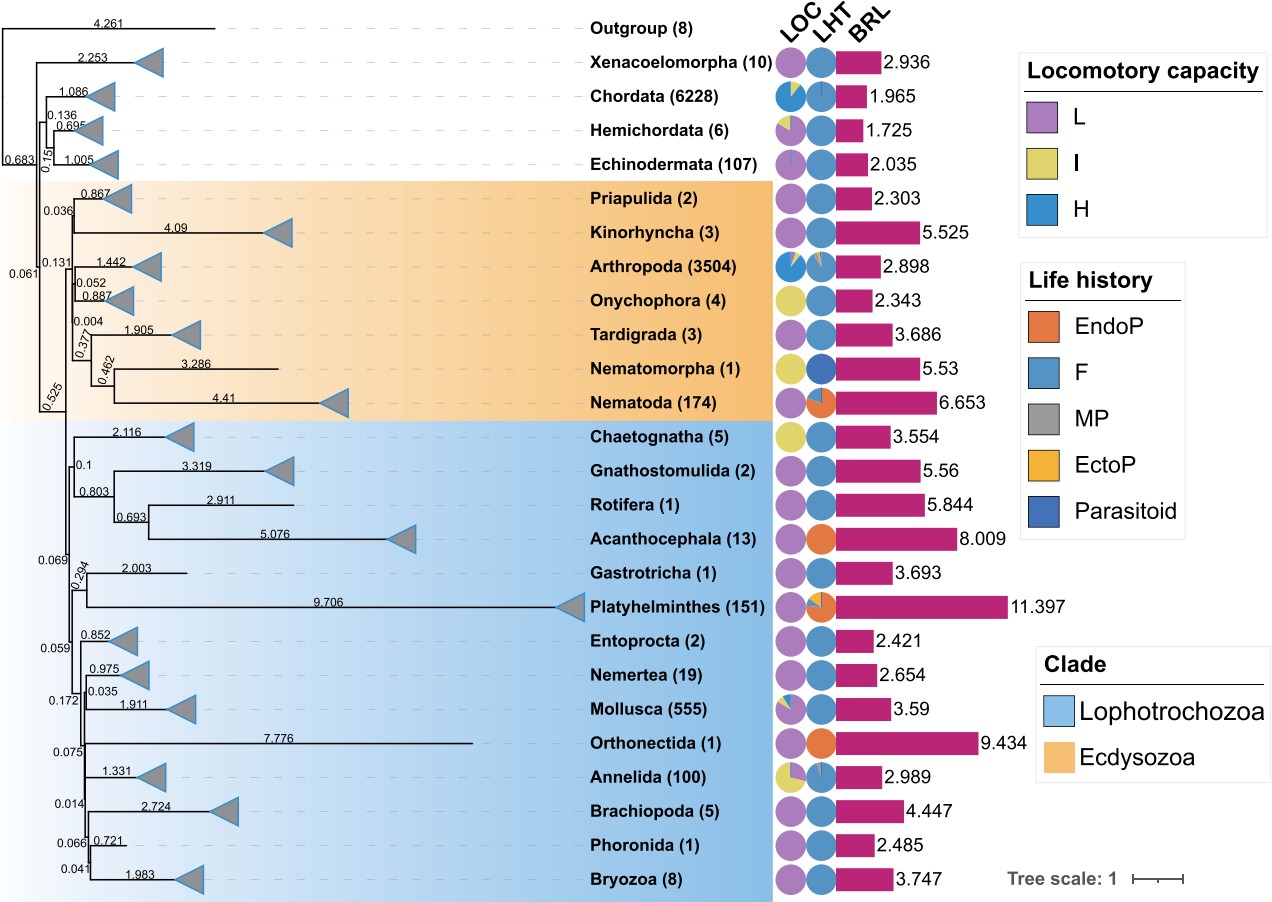

**Fig. 1 | Life history, locomotory capacity and average branch lengths of the Bilaterian dataset used to conduct the analyses.** The phylogeny was inferred using amino acid sequences of 12 mitochondrial protein-coding genes and constrained at the phylum level according to Laumer et al. [54]. Two major bilaterian clades, Ecdysozoa and Lophotrochozoa, are indicated by shading. A triangle at the branch tip indicates that the phylum was collapsed to a single branch. Leaf labels show phyla names and the number of species (in parentheses). Pie charts show the proportion of each life-history (LHT) and locomotory capacity (LOC) category in each phylum. Branch lengths (BRL) are shown in the phylogram, with the average root-to-tip branch lengths per phylum shown as bars to the right and exact numbers shown above the bars. Source data are provided as a Source Data file.

micropredators vs. free-living (>0.7) (Supplementary Table 2 and Supplementary Fig. 1). For five Nematoda (Strongyloididae) species, we had evidence that they alternate between the free-living and parasitic generations, so we excluded them from subsequent analyses. Endoparasitic and ectoparasitic species (macroparasites) were identified in five phyla (Fig. 1; Supplementary Table 3). Among these, all four phyla that contained large ratios of macroparasites had the highest average branch lengths (brl) in the bilaterian dataset: Platyhelminthes (brl = 11.4, 96% of parasitic species), Orthonectida (9.4, only one species, endoparasite *Intoshia linei*), Acanthocephala (8.0; 100% endoparasitic), and Nematoda (6.8, 80% parasitic). Arthropoda was the fifth phylum containing parasitic species; it did not exhibit exceptionally long branches (2.9), but the ratio of macroparasites was low (4.1%). Other phyla with average branch lengths more than twice the size of the average branch length calculated across all lineages (2.58 × 2 = 5.16) comprised Rotifera (5.8), Gnathostomulida (5.6), Nematomorpha (5.5) and Kinorhyncha (5.5) (additional data in Supplementary Tables 3 and 4, and Supplementary Note 1). Accounting for some uncertainty in evolutionary scenarios (e.g. in Nematoda and Platyhelminthes), independent origins of endoparasitism were observed in Acanthocephala (1 origin), Arthropoda (2: Pentastomida and Rhinonyssidae), Orthonectida (1), Platyhelminthes (2 or 3), and between 5 and 7 origins in Nematoda according to the evolutionary scenario proposed previously[34]. We identified 12 putative origins of ectoparasitism in Arthropoda and 1 in Platyhelminthes (see Supplementary Note 2 and

Supplementary Table 3). Endoparasites exhibited by far the longest average branches in the dataset (9.05), followed by ectoparasites (5.54), parasitoids (3.42), micropredators (2.50) and free-living species (2.37) (Fig. 2a, Supplementary Fig. 2). The two macroparasitic categories also exhibited by far the largest standard deviation values (Supplementary Table 2). Aside from the micropredator vs. free-living comparison, all differences were significant (exact $p$-values of Tukey HSD tests in Supplementary Data 1: Worksheet 2). Effect sizes were very large for all pairwise comparisons involving ectoparasites (Cohen's d > 0.8), including the comparison with endoparasites (1.3), and huge for all comparisons involving endoparasites (>2.3), aside from ectoparasites vs. endoparasites (1.3; Supplementary Fig. 1).

As regards the locomotory capacity categorisation, the vast majority of Bilateria were classified as 'High', but both remaining categories comprised over 900 species, ensuring a large sample. Low locomotory capacity was inferred as the ancestral state for Bilateria with 99.9% probability, so we inferred independent origins of high locomotory capacity in only three phyla (Fig. 1): 1) in the Cephalopoda class of Mollusca; 2) in Chordata; 3) in Arthropoda, which exhibited a complex phylogenetic distribution of different categories, requiring at least 15 High ↔ Low transitions. In agreement with our working hypothesis, branch lengths were the lowest in the High and Intermediate categories (2.27 and 2.30 respectively–small effect size), and more than twice longer in the Low locomotory capacity category (5.05–very large effect size; Fig. 2e and Supplementary Fig. 1).

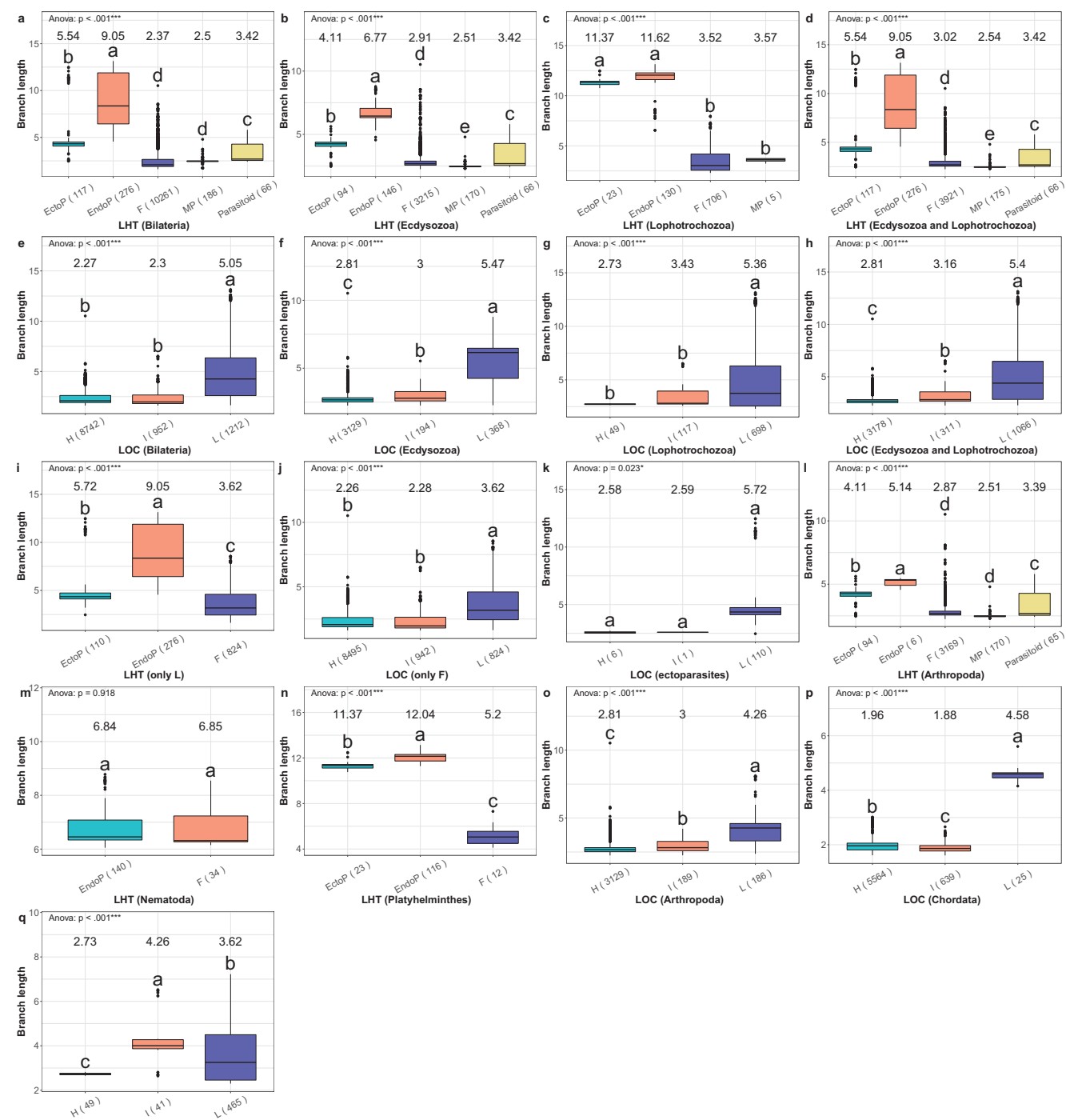

**Fig. 2 | Branch length comparison between different life history and locomotory capacity categories.** In each panel, the branch length distribution boxplot for each dataset is shown on the y-axis, while datasets compared are shown on the x-axis. The titles below the panels are presented in the following format: categorisation (dataset). Categorisations comprise life-history (LHT) and locomotory capacity (LOC). In the LHT categorisation, F is free-living, EndoP is endoparasitic, EctoP is ectoparasitic, and MP is micropredatory. In the LOC categorisation, H is high, I is intermediate, and L is low. Boxplots show the minimum, first quartile, median, third quartile, and maximum, plus outliers. Average branch length values are shown above the boxplots. ANOVA results are shown in the upper left corner. Different letters above the boxplots indicate statistically significant differences ($p < 0.05$; Tukey HSD). The number of species included in the analysis is shown next to the category name (x-axis). Source data are provided as a Source Data file.

Using the phylogenetic generalised least squares (PGLS) ANOVA test, we found significant differences in all comparisons (pairwise and overall) and all datasets, regardless of how small the differences were (Supplementary Fig. 3), so we suspect that it overestimates the statistical significance of differences, and we relied on Tukey HSD tests in the manuscript unless specified otherwise. We attempted to assess whether these relationships are stable when outliers are removed, but this step removed all endoparasites, which indicates that it removed valuable signal from the dataset. Despite this, the full-dataset patterns were mostly upheld in the life history analyses, and partially in the locomotory capacity analyses (Supplementary Fig. 4). We further assessed whether these patterns (inferred across all Bilateria) are

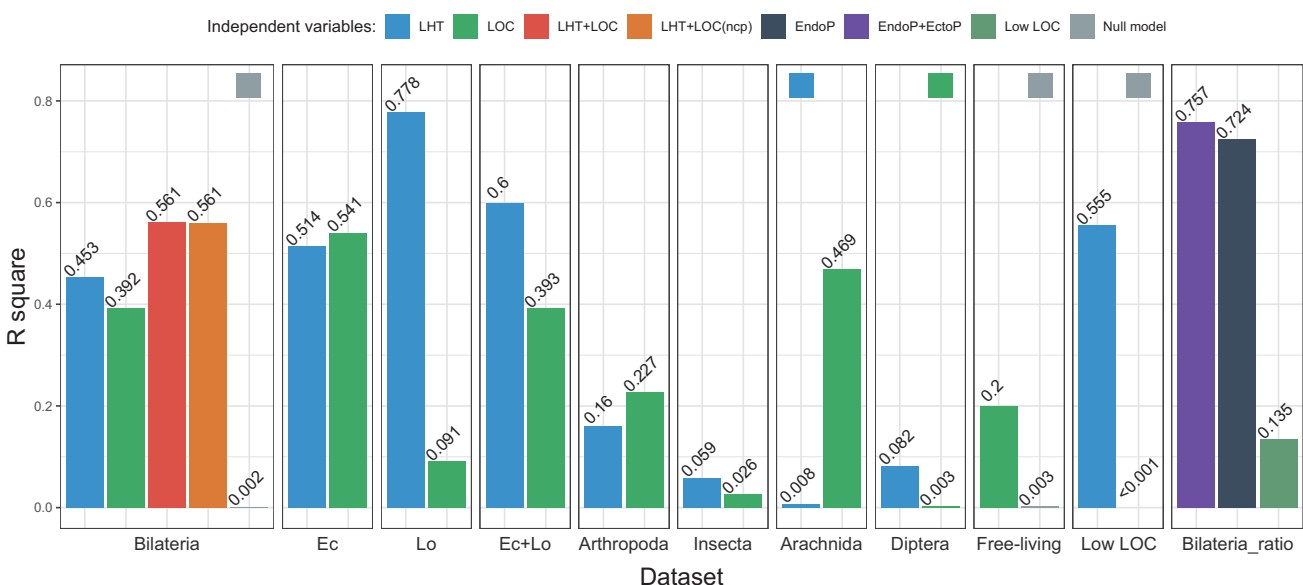

**Fig. 3 | Phylogeny-corrected *lmekin* multilevel regression analyses with branch length as the dependent variable.** The y-axis shows the proportion of branch length variation explained by the independent variable (adjusted R²). The independent variables were: life history categorisation (LHT), locomotory capacity categorisation (LOC), or both together (LHT + LOC). Except for the LHT + LOC(ncp), all other models were corrected for phylogeny. EndoP is the ratio of endoparasites, EndoP+EctoP is the ratio of endoparasites+ectoparasites, and Low LOC is the ratio of taxa assigned to the Low locomotory capacity category, across phyla. Ec Ecdysozoa, Lo Lophotrochozoa. In cases where bars are very small, to discern what they represent, we have shown the legend above the bar. Source data are provided as a Source Data file.

consistent at lower phylogenetic levels, but analyses were hampered by the fact that many lineages did not comprise all relevant life history and locomotory capacity categories, as well as by small numbers of samples in many cases. Concerning our major hypotheses that macroparasitic/low locomotory capacity lineages are expected to have longer branches than free-living ones and those with high locomotory capacity, patterns at lower taxonomic levels mostly supported them, with two major exceptions: 1. Nematoda, where branch lengths of endoparasitic lineages were only nonsignificantly longer than in the free-living (this pattern was also observed at lower taxonomic levels: class and order); and 2. Arachnida, where ectoparasites had nonsignificantly shorter branches than the free-living ones. In both cases, the statistical power of the analyses was very weak for key comparisons. Two additional minor exceptions comprised Diptera and Hexanauplia, but in both cases key categories had only three representatives, which also caused low statistical power (further details in Discussion, Supplementary Notes 3–5, and Supplementary Figs. 5–11).

As parasitism and locomotory capacity are partially overlapping variables, we attempted to discern their impacts using further subsets of data. To reduce the effect of locomotory capacity variability, we focused only on the Low locomotory capacity category and divided it along the life history lines. The pattern produced by the overall bilaterian dataset was upheld, but the average branch lengths were increased compared to the overall dataset from 5.54 to 5.72 in ectoparasites and from 2.37 to 3.62 in free-living species (Fig. 2a, i). As all endoparasites were classified in the Low locomotory capacity category, their branch length was unaffected (9.05). To remove the effect of parasitism on the locomotory capacity classification, we conducted analyses using only the free-living species. Again, the pattern was upheld, with branch lengths almost unchanged in the High (2.26) and Intermediate (2.28) categories, but much shorter in the Low locomotory capacity category (3.62 vs. 5.05) (Fig. 2e, j). Notably, within this dataset (>10,000 species), among the 200 species with the longest branches, 182 were classified in the Low locomotory capacity category, and only 10 in the High category. We further focused on the

ectoparasitic dataset and compared lineages according to the locomotory capacity categorisation: species in the High and Intermediate categories (all Arthropoda: Insecta) had almost twice shorter branches than those in the Low category, but this analysis was hampered by a small number of samples in the former two categories (Fig. 2k). Moreover, branch lengths of ectoparasites with high/intermediate locomotory capacity did not differ significantly from those of free-living Insecta (Supplementary Fig. 12). Finally, to make sure that our findings are not skewed by differing evolutionary rates among mitochondrial genes, we also confirmed that topologies inferred using individual protein-coding genes produce congruent results (Supplementary Note 6 and Supplementary Fig. 13).

## Multiple regression and LRT analyses
The *lmekin* multilevel phylogenetic regression analysis of the bilaterian dataset divided into five life history categories and three locomotory capacity categories found that these two categorisations together explained 56% of the branch length variability in the dataset. Individually, life history explained 45% and locomotory capacity 39% of variability (Fig. 3). We tested whether dividing the dataset differently along the life history lines (e.g. EndoP+EctoP vs. all other categories) affects the results, but R² values were largely unaffected (Supplementary Table 6). Removing the phylogenetic signal by running the analysis without importing the phylogenetic matrix produced a negligible difference in R² values (Fig. 3). We also confirmed these results using *brms*: conclusions were unaffected, with a minor difference that R² values were on average 2–4% lower than those inferred by *lmekin* (Supplementary Table 7; Supplementary Data 1: Worksheet 3). To reduce collinearity between the parasitism and Low locomotory capacity category, we first removed the effect of life history categorisation (parasitism) by focusing only on the free-living dataset: in this dataset, locomotory capacity categorisation explained 20% of the variance in branch lengths (Fig. 3). To remove the effect of locomotory capacity variability, we conducted an analysis using only the Low locomotory capacity category: life history categorisation explained over 55% of the branch length variance. We designed likelihood-ratio

**Table 1 | Likelihood Ratio Tests (LRT) designed to assess the impact of life history and locomotory capacity on the branch length**

| model | Ind Var | formula | $R^2$ | DF | Log_lik | AIC | P |
|---|---|---|---|---|---|---|---|
| **Bilateria** | | | | | | | |
| Alternative | LOC | brl ~ LOC + RE | 0.3689 | 5 | −12964 | 25938 | 0 |
| Null | | brl ~ RE | 0.0015 | 3 | −15466 | 30939 | |
| Alternative | LHT | brl ~ LHT + RE | 0.4262 | 7 | −12445 | 24905 | 0 |
| Null | | brl ~ RE | 0.0015 | 3 | −15466 | 30939 | |
| Alternative | LHT + LOC | brl ~ LHT + LOC + RE | 0.5279 | 9 | −11381 | 22781 | 0 |
| Null | | brl ~ RE | 0.0015 | 3 | −15466 | 30939 | |
| Alternative | LHT + LOC | brl ~ LHT + LOC + RE | 0.5279 | 9 | −11381 | 22781 | 0 |
| Null | | brl ~ LHT + RE | 0.4262 | 7 | −12445 | 24905 | |
| Alternative | LHT + LOC | brl ~ LHT + LOC + RE | 0.5279 | 9 | −11381 | 22781 | 0 |
| Null | | brl ~ LOC + RE | 0.3689 | 5 | −12964 | 25939 | |
| **Free-living** | | | | | | | |
| Alternative | LOC | brl ~ LOC + RE | 0.1886 | 5 | −13487 | 26984 | 0 |
| Null | | brl ~ RE | 0.0027 | 3 | −14546 | 29097 | |
| **Low LOC** | | | | | | | |
| Alternative | LHT | brl ~ LHT + RE | 0.5223 | 6 | −1272 | 2555 | 0 |
| Null | | brl ~ RE | −6.6E-06 | 3 | −1719 | 3445 | |

The analyses were designed to test the alternative hypothesis that the independent variable (Ind Var; fixed effect) has a significant effect on the branch length (brl; the dependent variable in all analyses) once the impact of other variables is controlled for (Null hypothesis). Locomotory capacity (LOC) and life history (LHT) were tested as independent variables in different combinations. Analyses were conducted on the overall dataset (comprising all Bilateria species), and two sub-datasets: free-living and Low LOC categories. $R^2$ represents the proportion of the variance in the dependent variable that is explained by the independent variable in a linear regression model. RE is a random effect, DF is degrees of freedom, Log_lik is Log-likelihood, and AIC is the Akaike Information Criterion (smaller value = better fit of the model). P is the P-value for the alternative hypothesis that the independent variable has a significant effect on the branch length, inferred using a chi-square test in R. All models were corrected for phylogeny.

tests (LRT) to assess the explanatory power of life history and locomotory capacity categorisations in relation to the branch lengths. In all cases, these categorisations had a better model fit (lower AIC value) and greater explanatory power (higher $R^2$ value) than the null model (includes only random effects and correction for phylogeny), and both variables together performed better on both accounts than either of the variables alone (Table 1).

To test for lineage-specific effects, we conducted these same analyses on all major taxonomic categories that contained a mix of relevant life history and locomotory capacity categories. In the Ecdysozoa, the life history and locomotory capacity categorisations explained similar proportions of the variance: 51% and 54% respectively (Fig. 3). Lophotrochozoa exhibited a different pattern: life history explained a major proportion of the variance (78%), whereas locomotory capacity explained a small amount of it (9%), but this dataset contained a tiny proportion of High locomotory capacity lineages. The Ecdysozoa+Lophotrochozoa dataset produced an intermediate result. In Arthropoda, locomotory capacity explained a greater (23%) proportion of the variability than life history (16%) categorisation. At lower taxonomic levels, both variables explained relatively small proportions of the variability (<9%), with the exception of Arachnida, where locomotory capacity explained 47% (Fig. 3). These analyses were hampered by small numbers of samples and a lack of diversity in studied categories. The ratio of endoparasites in a phylum explained 72% of the branch length variance across the Bilateria, with the addition of ectoparasites having a comparatively small impact on the explanatory power (76%). The ratio of species assigned to the Low locomotory capacity group explained 13.5% of the variance (Fig. 3).

## Selection pressure analyses

To attempt to elucidate the contributions of different selection pressures on elevated branch lengths, we further tested two hypotheses: 1) parasitic and Low locomotory capacity lineages exhibit signals of relaxed purifying selection pressures; 2) parasitic lineages exhibit elevated rates of directional evolution. These analyses had to be conducted on strongly reduced datasets due to computational limitations and the existence of multiple genetic codes within the Bilateria, so we separately tested a subset of invertebrates and all available Platyhelminthes (further details about datasets and results in Supplementary Note 7 and Supplementary Table 8). Generally, purifying selection pressures analyses were in agreement with our working hypotheses and branch length patterns: relaxed in endoparasites and ectoparasites compared to the free-living, but there were some exceptions. Contrary to our hypotheses, endoparasites exhibited significantly stronger purifying selection pressures than ectoparasites in the invertebrate dataset, but this analysis encountered multiple convergence problems, so results should be interpreted with care. Parasitoids exhibited highly significantly relaxed selection pressures in comparison to free-living taxa. Comparisons of High and Low locomotory capacity categories produced similar ω values and inconsistent results, but our hypotheses were supported in Arthropoda. Apart from flatworms, there was no consistent evidence for increased levels of directional selection in parasitic compared to free-living lineages. Similarly, there was no evidence for different levels of directional selection between the Low and High locomotory capacity categories.

## Discussion

We showed herein that previous evidence that parasitic lineages exhibit elevated mitogenomic evolutionary rates in comparison to free-living lineages is largely limited to selected Arthropoda lineages and previous results were often inconsistent. We hypothesised that locomotory capacity may be the key factor driving these accelerated evolutionary rates. By using almost 11,000 currently available bilaterian mitogenomes, we provide statistical evidence demonstrating that significantly elevated mitogenomic evolutionary rates in endoparasitic and most ectoparasitic lineages are an almost universal phenomenon in Bilateria. The effect of endoparasitism on evolutionary rates was particularly strongly pronounced. We also found support for the hypothesis that evolutionary rates are higher in bilaterians with a strongly reduced locomotory capacity, but the effect was less prominent. Notably, the top four phyla displaying disproportionately elevated branch lengths were predominantly parasitic: Platyhelminthes, Orthonectida, Acanthocephala and Nematoda.

Among the four exceptions in relation to our working hypotheses, locomotory capacity may at least partially provide an explanation for these outliers in three of them. We found that branch lengths of endoparasitic Nematoda lineages were only nonsignificantly higher than those of the free-living. All Nematoda possess low locomotory capacity, and they have an evolutionary complex mix of parasitic and free-living life histories, with relatively common major life history switches[34]. In addition, many plant-parasitic nematodes resemble endoparasitic ones in many aspects, but according to the definition of parasitism that we relied on in this study[28], they were classified as free-living. Combined, these factors may explain this major outlier. In Arachnida, ectoparasitic lineages (ticks) had shorter branches than the free-living ones, but this was a product of exceptionally long branches in free-living mites, which possess merely a rudimentary locomotory capacity, comparable to that of parasitic lineages. Overall, locomotory capacity was a very strong predictor ($R^2 = 47\%$) of branch length in Arachnida. Dipteran ectoparasites (Hippoboscoidea) on average have far greater locomotory capacity than most other ectoparasites, so locomotory capacity also appears to be a suitable explanation for the absence of significantly elevated evolutionary rates in ectoparasitic Diptera. The order Sessilia of the Hexanauplia class was previously

recognised as a major outlier among crustaceans: despite its sessile lifestyle, it appears to exhibit very strong purifying selection pressures[10]. This explains the shortest branches in lineages with low locomotory capacity in Hexanauplia and indicates that Sessilia is a major outlier not only among crustaceans but also among all bilaterian animals (more details in Supplementary Notes 4 and 5).

Combined, these findings support the hypothesis that the reduction of locomotory capacity is among the key variables behind the elevated evolutionary rates in parasitic lineages. However, the proportion of variance explained by the locomotory capacity categorisation was on average smaller, and in some cases much smaller, than the proportion of variance explained by the life history categorisation, which indicates that locomotory capacity is not the sole explanatory variable for the increased evolutionary rates in parasitic lineages. In certain cases, the limited explanatory power can be attributed to the low variability in locomotory capacity. For example, in Lophotrochozoa, where the only lineage exhibiting high locomotory capacity was Cephalopoda. In other cases, the impact of locomotory capacity appears to be inconsistent.

When the effect of parasitism was removed, the proportion of variability in branch length across the free-living Bilateria explained by locomotory capacity decreased from 39% to 20%, which is still a remarkably good predictive power. Additional analyses provided further support for this observation. For example, parasitic life histories characterised by comparatively high locomotory capacity, comprising a subset of ectoparasites (fleas) and parasitoids, exhibited significantly lower evolutionary rates compared to parasites with low locomotory capacity. Interestingly, ectoparasites with high locomotory capacity displayed branch lengths similar to those of free-living species possessing high locomotory capacity. Moreover, the majority (approximately 90%) of extreme branch lengths within the free-living dataset were found in species with low locomotory capacity. The selection pressure for high locomotory capacity may also explain the much smaller effect of parasitism and inconsistent results observed in previous studies of parasitoid insects[6,15]. While these findings indicate that locomotory capacity is a major variable driving the heterogeneity in evolutionary rates among bilaterian animals, as well as elevated rates in parasitic lineages, we were unable to confirm the hypothesis that it is the most important variable. Other variables also appear to contribute to the elevated evolutionary rate in parasites, and in some cases may even outweigh the impact of locomotory capacity on mitogenomic evolution.

Among these might be the high dependence on the host, which allows a reduction of various metabolic functions in some parasitic lineages[29,30]. This reduction, in turn, allows for a relaxation of purifying selection pressures. Other factors comprise generation time, replication and repair machinery, $N_e$, directional selection driven by host-parasite arms race[12,35], etc. Our selection pressure analyses did not support the hypothesis of directional selection, but they consistently supported the hypothesis of relaxed purifying selection pressures in parasitic lineages. Locomotory capacity appeared to be a good predictor of this selection pressure relaxation in most, but not all, lineages, which indicates that other factors may also contribute to this phenomenon. In partial agreement with these findings, although plants do not exhibit variability in locomotory capacity across different life history strategies, elevated evolutionary rates have been proposed for all three genomes (nucleus, mitochondrion, and chloroplast) in parasitic plants[31]. The association between the metabolic dependence on the host and the strength of purifying selection was invoked as an explanation for this observation[31], but a follow-up study failed to find evidence for elevated evolutionary rates in plant mitogenomes[36]. Notably, there is also evidence of a reduction in size of nuclear genome in parasitic animals[33], but it remains unknown whether these increased mitogenomic evolutionary rates are also mirrored in nuclear genomes of animals. It is also worth noting that both parasitism and locomotory capacity may be associated with some of the aforementioned

variables, such as $N_e$. While parasitism may cause elevated evolutionary rates by reducing the $N_e$ through founder effects and increased speciation[6,14,15,37], locomotory capacity may achieve this by affecting (female) dispersal ability. However, the magnitude of this effect may depend on multiple ecological factors (e.g. it might be less pronounced in aquatic animals with a planktonic larval stage), resulting in inconsistent effects of parasitism.

Due to its much broader dataset scope, the resolution of our study is much higher than in previous attempts to tackle this question[6–10], but several caveats should be considered. A potential source of noise in our analyses lies in the presence of free-living stages in many nominally parasitic species[16]. Additionally, the boundaries among different parasitic life history categories, as well as nominally free-living (such as plant-parasites) and parasitic species[29], are often ambiguous. Moreover, due to the absence of clear criteria for categorizing species on the basis of their locomotory capacity, which is a continuous trait, there is some overlap between the Intermediate locomotory capacity category and the other two categories. Importantly, the introduction of the Intermediate category allowed us to avoid overlap between the High and Low locomotory capacity categories, enabling us to focus our tests on these two categories. As this categorisation of locomotory capacity was insufficiently precise to capture the full range of differences between bilaterian species, it is likely that the explanatory power of this variable was underestimated in our study. The selection analyses were weakened by the need to significantly reduce most datasets and the inherent challenges of inferring nucleotide homology in a deeply divergent dataset as ours. This may explain some inconsistent findings, such as the different patterns observed between the bilaterian and arthropod datasets in tests comparing Low and High locomotory capacity, where the Low category showed neutral selection and the High category exhibited relaxed purifying selection (Supplementary Table 8). Finally, our analyses did not include several variables that have been associated with mitogenomic evolution by previous studies, such as generation time, longevity, body size/mass, effective population size, metabolic rate, etc. (see Supplementary Information). However, the impacts of these competing variables were inconsistent across different lineages, especially when invertebrates were included[1,10,38,39]. Moreover, the effect sizes of these variables were generally much smaller than the ones observed in our study when a broad range of lineages was included. For example, the generation time and/or longevity explained about 12-13% of the variability in mitogenomic evolutionary rates in both vertebrates[2] and invertebrates[40], while body size explained only 6% in mammals (only *cytb* was used in this study)[2] (for further discussion of limitations please see Supplementary Information).

In conclusion, our analyses support previous findings that there is no single factor with the power to predict the patterns of mitogenomic evolution across all animal lineages[1,38,39]. However, across bilaterian animals, we found that parasitism explained approximately 45% of the variability, while locomotory capacity explained around 39%. Together, these two factors accounted for approximately 56% of the total variability in evolutionary rates of mitochondrial protein-coding genes at the amino acid sequence level. This indicates that we have identified the major explanatory variables for mitogenomic evolution in bilaterian animals. However, since locomotory capacity did not account for all the variability across different life history categories, it appears that multiple factors are underlying the elevated evolutionary rates in parasitic lineages. Furthermore, our findings indicate that in certain isolated lineages, other variables may have a greater impact on mitogenomic evolution than both parasitism and locomotory capacity.

## Methods
### Dataset
When we accessed the data (10th March 2022), there were 11,284 animal and 11,017 bilaterian mitogenomes available in the RefSeq

database. The 'standard' metazoan (animal) mitogenome is a commonly a circular molecule ≈15 Kbp in size that contains 37 genes: 13 protein-coding genes (PCGs), 2 rRNA genes and 22 tRNA genes, but there are major deviations from this architecture in some lineages. While some deviations from this canon have been observed in isolated bilaterian lineages, almost all of the major discrepancies in the protein-coding gene content map to the non-bilaterian metazoans[41]. As this would complicate some comparative analyses, and as non-bilaterian mitogenomes represented only about 2% of the total available Animalia dataset, in this study we focused only on Bilateria. PhyloSuite[42,43] was used to standardise and extract the mitogenomic data, as well as generate comparative tables for the dataset. After the removal of all unannotated mitogenomes, most hybrids between species, and identical mitogenomes (we suspected species misidentification in these cases), the final dataset comprised 10,914 species (+8 outgroups).

Many mitogenomes had duplicated genes. As only one gene could be kept for subsequent analyses, we devised a pipeline that resolved the duplicates. The pipeline first searches for the existence of stop codons in duplicates, and then removes the one possessing them, as this implies non-functionality. If this step does not resolve the duplicates, then it compares the orthologous genes of the lowest available identical taxon in the dataset, and keeps the most conserved duplicate. This function was added to the latest version of PhyloSuite (v1.2.3)[43].

## Categorisation of life-history strategies
The dataset was classified according to life history into five categories: endoparasites (EndoP), ectoparasites (EctoP), parasitoids, micropredators (MP), and free-living (F). Endoparasites comprised parasites living inside the host's body, which included intestines, nasal passages, etc. (so this category includes mesoparasites as well) (Supplementary Methods). Ectoparasites attach themselves or permanently live on the outside of the host's body. As we expected endoparasites to have a lower locomotory capacity and higher metabolic dependence on the host, we expected their evolutionary rates to be higher than those of ectoparasites. Parasitoids include organisms that are parasitic during a part of their life cycle; often parasitoids have a parasitic larva stage that eventually kills its host, followed by a free-living adult stage (Supplementary Methods). We used parasitoids to test our hypothesis that parasites with high locomotory capacity and incomplete metabolic reliance on the host should not exhibit elevated evolutionary rates. To further confirm our predictions, we separately classified micropredators. These organisms resemble many ectoparasites in their haematophagous feeding habits, but they only visit the host for feeding, and otherwise they are free-living (e.g., mosquitoes). We hypothesised that their evolutionary rates should not differ from other free-living organisms. Eight species were left out from statistical analyses because their life history classification was unclear: for three species we could not find data, and five Strongyloididae (Nematoda) species had a life cycle that alternates between free-living and parasitic generations[44,45] (details in Supplementary Methods). All other bilaterian species were classified as free-living.

## Categorisation of locomotory capacity
As we failed to find a suitable categorisation of locomotory capacity in animals, we adopted the 'visual interaction hypothesis', which is the best explanatory variable for variability in scaling coefficients between the mass-specific metabolic rate in marine animals: high metabolic demand follows strong selection for locomotory capacity for pursuit and evasion in visual prey/predators inhabiting well-lit oceanic waters, whereas limited visibility allows for reduced locomotory capacity, reflected in low metabolic rates[46,47]. We divided the dataset into three locomotory capacity categories. 1. High (H), comprising all species expected to rely on locomotion for pursuit and evasion of prey/predators. This category comprised a vast majority of species. 2. Low (L), comprising all species that have merely a rudimentary locomotory

capacity. This comprises sedentary species, or those capable only of minimal, very slow locomotion, that do not rely on locomotion for pursuit and evasion of prey/predators. This category comprised 1212 species, mostly from lineages such as nematodes, flatworms, sessile molluscs and crustaceans, parasitic crustaceans, ticks, some annelids, mites, Acanthocephala, Diplura, Brachiopoda, Bryozoa, Ascidiacea, Echinodermata, Entoprocta, Gastrotricha, Gnathostomulida, Hemichordata, Kinorhyncha, Nemertea, Onychophora, Orthonectida, Phoronida, Priapulida, Tardigrada, and Xenacoelomorpha. 3. Because locomotory capacity is a continuous trait, it is impossible to categorise species into only two categories without creating a large overlap between the two categories. Therefore, to minimise the overlap between the Low and High locomotory capacity categories, we designed a third category: Intermediate locomotory capacity (I). This category comprises species that would be expected to possess more than a rudimentary locomotory capacity, but also rely on strategies other than locomotion to evade/pursue predators/prey. Examples are certain flightless insects and Collembola, certain bathyal, abyssal and stygobitic crustacean and fish lineages, Chaetognatha, certain amphibian and reptilian lineages, some mammals, some Bivalvia and Cephalopoda, some Gastropoda, Nematomorpha, and Rotifera (further details in Supplementary Information: Methods and Supplementary Data 1: Worksheet 1).

## Statistical analyses
For pairwise comparisons of branch lengths between different groups, we conducted Tukey HSD tests using the R package agricolae (v1.3.5). As our data violated the assumption of independence due to varying levels of phylogenetic relatedness, we further addressed this problem by breaking down the dataset into pairs of categories and conducted PGLS ANOVA tests using the *nlme* package v3.1.152[48]. We also conducted these tests on the overall dataset. To assess the relative impacts of different variables on branch length, we used two algorithms designed to account for the phylogenetic relatedness of data: linear fixed-effect models accounting for kinship implemented in the *lmekin* function in *coxme* v2.2.16[49], and phylogenetic multilevel Bayesian models implemented in *brms* v2.16.1[50]. For both analyses, we used a matrix of phylogenetic distances extracted from the phylogenetic tree using the *ape* v5.5 package, and log-transformed the branch length data to reduce the nonnormality of distribution. The $R^2$ parameter evaluates the proportion of the variance in the dependent variable explained by the independent variable. The $R^2$ value of *lmekin* models was calculated using the "r.squaredLR" method available in the MuMIn v1.46.0 package in R, and Bayesian $R^2$ was inferred using the *bayes_r2* function of *brms* package in R. The AIC value of each *lmekin* model was calculated using the "AIC" method in R. The LRT test of *lmekin* models was conducted using the $\chi^2$ test following this tutorial: https://aeolister.wordpress.com/2016/07/07/likelihood-ratio-test-for-lmekin/ (version: 7/7/2016; last accessed 30/3/2023). Branch length outliers in the dataset were defined as "1.5 times the interquartile range" and identified using the "boxplot" function in R.

## Phylogenetic analyses
The phylogeny of Bilateria was inferred using the dataset comprising 10,914 bilaterian mitogenomes and 8 non-bilaterian Animalia outgroup species. All steps of phylogenetic analyses were conducted using PhyloSuite and its plug-in programs. We conducted several analyses to assess the stability of root-to-tip branch lengths of individual leaves (see Supplementary Methods for details). Correlations between pairs of trees were all greater than 96%, which indicates that this variability had a small impact on our downstream statistical analyses (Supplementary Data 1: Worksheet 4). To conduct these, we used a phylogram constructed using concatenated amino acid sequences of 12 mitochondrial protein-coding genes (*atp8* was removed because it was missing from multiple lineages), and mtZOA+G4 + F + C50 model

in combination with the approximate Bayes test and "--fast" parameter in IQ-TREE v2.2.0.7.mix[51]. All sequences were aligned using MAFFT v7.475[52]. Alignments were trimmed by trimAl v1.2rev59[53], and concatenated by PhyloSuite. The phylum-level tree topology was constrained according to Laumer et al. [54]. The molecular evolution rate was defined as the root-to-tip branch lengths[4,55], which were extracted using the TreeSuite function in PhyloSuite. The Maximum Likelihood method in BayesTraits v4.0.1[56] was used to infer the ancestral states of traits (further details in Supplementary Methods).

### Selection pressure analyses
To study selection pressure patterns in the dataset, we used two tools from the HYPHY suite v2.5.42[57]. RELAX was used to test the hypothesis that selection pressures are relaxed in a selected subset of branches against the background branches[58]. BUSTED was used to test the hypothesis that directional selection is associated with a certain phenotype (test branches)[59]. As these tests require a single codon table, and codes vary across the bilaterian dataset, it had to be divided along the genetic code lines. Some of the datasets had to be further reduced to make them computationally feasible. We did this by selecting closely related test and background branches across the phylogenetic tree (details in Supplementary Methods).

### Reporting summary
Further information on research design is available in the Nature Portfolio Reporting Summary linked to this article.

## Data availability
All data used in this study were retrieved from the NCBI's GenBank RefSeq database; the full list of Accession Numbers is provided in Supplementary Data 1: Worksheet 1. Source data are provided with this paper.

## Code availability
The code written for filtering the mitogenome data was subsequently incorporated into PhyloSuite v1.2.3[43], available from https://github.com/dongzhang0725/PhyloSuite or https://pypi.org/project/PhyloSuite/. The remaining code written for this study was deposited in Zenodo: https://doi.org/10.5281/zenodo.7940125.

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

## Acknowledgements

This work was supported by the National Natural Science Foundation of China [grant numbers 32102840 (D.Z.) and 32360927 (D.Z.)]; the Key Project of Natural Science Foundation of Tibet [XZ202301ZR0028G (D.Z.)]; the National Natural Science Foundation of China [grant numbers 32230109 (key program, G.T.W.), and 31872604 (H.Zou)]; and the Start-up Funds of Introduced Talent in Lanzhou University [561120206 (D.Z.)]. The computation works were supported by the Big Data Computing Platform for Western Ecological Environment and Regional Development, and the Supercomputing Center of Lanzhou University. The funders had no role in study design, data collection and analysis, decision to publish, or preparation of the manuscript. We would like to thank Prof. Jianquan Liu, Dr. Sergei L. Kosakovsky Pond and Prof. Chuan Yan for useful discussions and technical help.

## Author contributions

I.J. and D.Z. conceived the study. I.J., D.Z., H.Zh, and X.L. designed the analyses. D.Z., I.J., C.Y.X., and H.Zou were involved in the acquisition of data. D.Z., C.Y.X., I.J., and H.Zou performed the bioinformatic and statistical evolutionary genomic analyses. D.Z. developed and wrote the new code needed to conduct the analyses. I.J. drafted the manuscript. D.Z. and G.T.W. revised the manuscript and supervised the project.

## Competing interests

The authors declare no competing interest.
