## [Peer Review File · Nature Communications]

REVIEWER COMMENTS

Reviewer #1 (Remarks to the Author):

1. Overall, I found this to be an interesting and relevant study of the role of two important traits (parasitism and locomotory ability) in predicting the rate of molecular evolution in animal mitochondrial genomes. In general, I found the study to be clearly written and convincingly conducted, with a large and expansive sample size and sets of analyses (both phylogenetic and non-phylogenetic). I have a few comments that I believe will help the authors to improve their paper.

2. One of my more major comments relates to investigations and reporting upon data quality. In large, public databases, I frequently find errors, such as bacterial contaminant sequences labeled at animals. Given that such types of errors can lead to long branch lengths, and erroneous branch length estimates could have a major impact on conclusions for this study, it is important to clarify briefly (in the main manuscript) how data quality was scrutinized. For example, I suggest to look at distance matrices for each individual gene and investigate outliers (e.g. through BLAST and trees). It can also be clarified if all alignments for protein-coding genes were checked for stop codons.

3. I suggest that it's important and interesting to discuss the potential implications of this work beyond the mitogenome. Would you expect patterns to be similar in the nuclear genome? In the discussion, I also suggest that variation in locomotory ability or dispersal distance by sex are relevant to consider, in light of the (mostly) maternal inheritance of the mitogenome.

4. The conclusions would be more convincing if additional possible major covariates of molecular rate variability from the literature (e.g. body size) were included explicitly in the multivariable analyses, or, at the least, considered in turn in the discussion. Other major traits can vary across phyla. Body size has been shown to be a significant correlate of rates in birds, for example, and this trait may be especially relevant given that the majority of the data analyzed are Chordata. Benoit Nabholz, Robert Lanfear, Jérôme Fuchs. Body mass-corrected molecular rate for bird mitochondrial DNA. *Molecular Ecology*, 2016, 25 (18), pp.4438 - 4449. [ff10.1111/mec.13780](https://doi.org/10.1111/mec.13780)
ffhal-01387904f

5. typo. Intro. Line 73. "section" should apparently read "selection".

6. Methods. I would like to see more detail about the classification of the organisms into locomotory capacity groupings, as that is a tricky issue. As well, as part of full reproducibility of the work, I would like to request that the authors include the biological trait data clearly, such as a supplementary file available for download in a standard format (e.g. CSV), with the trait classifications available for all species analyzed.

7. Code. The authors mention writing custom code for these analyses. As part of transparency/reproducibility, I would recommend publishing the code as a supplementary file and/or publicly on a well-known site (e.g. on GitHub).

8. Line 154-155. What phylogenies were used to count the number of evolutionarily independent transitions to parasitism? And what method was used for the trait state mapping? Maximum parsimony mapping? Or ML or a different method?

9. Line 160. Typo. "braches" should be "branches"

10. Line 191. The authors refer to the number of species in the locomotory groups and state there is good statistical power. However, the number of evolutionarily independent transitions between locomotory groups is more relevant here than the absolute number of species and should be reported.

11. I would find Figure 1 to be more informative with regards to this study if the tips were colour coded in a different way... perhaps the tips could be pies or bars with the proportion of species studied in each phylum with each trait. Or, the tree could be presented side by side with the trait information. (Such a plot can be made in various ways, but one option to consider is to create an informative phylogeny + trait figure using the R package ggtree.)

12. The version number and date of access should be given when citing an online tutorial.

13. Clarify the definition of used to identify statistical outliers. Also, were single-gene trees investigated for outliers in addition to the overall tree?

14. The phylogenetic level at which the substantial explanatory power was found needs to be mentioned in the abstract. Some other traits have been shown to have very strong explanatory power within some groups. Therefore, it is important to clarify up front the nature of the findings and the level at which they apply. I suggest it is important not to accidentally imply that rates don't vary a lot within some groups in association with traits.

15. I would find it easier to draw conclusions from Table 2 if the results were presented in figure format (with the full information kept as either table or supplementary file). For example, perhaps the taxa could be along the x-axis, and the y-axis could show the p-values or R2 values. The different variables (and combination of variables) could be shown as symbol/colour combinations. It would be easier to draw a conclusion about the relative explanatory strength of the variables, as well as the consistency of the results across taxa, if the information were to be presented in visual format.

Again, I enjoyed reading this paper, and I hope that the authors and editor will consider my comments for improving the research further.

Reviewer #2 (Remarks to the Author):

In this interesting manuscript, Ivan Jakovlić and colleagues study the determinants of the evolution of the proteins encoded by the mitochondrial genome of Bilaterians. The underlying logic of the authors is that the mitogenome plays a central role in energy production thanks to its coding capacities for 13 proteins involved in the respiratory chain (CYB, COXI-II-III, NAD1-2-3-4-4L-5-6, and ATP6-8). Consequently, the strength of purifying selection acting at the DNA level on mitochondrial genes should be positively correlated with the energy expenditure of the organisms. The latter would in turn produce a positive correlation with the strength of selection for locomotory capacity.

To test these hypotheses, the authors assemble a dataset of life history traits (parasitic versus free-living), locomotory capacity traits, and mitochondrial proteins for 10,911 bilaterian taxa. From the mitochondrial alignments, they estimate branch lengths (BL) on a reference phylogenetic tree. Their statistical analyses evaluate the degree of correlation between these BL and the life history and locomotory capacity traits. Their main finding is that the proportion of BL variance explained by the locomotory traits is on average (much) smaller than the proportion of BL variance explained by the life history traits. Therefore, the locomotory capacity is not the only explanatory variable for elevated evolutionary rates in parasitic lineages. Of note, parasitic taxa possessing a free-living (adult) stage are evolving under selection for higher locomotory capacity. Importantly, life history categorisation (parasitism) explained up to 37%, locomotory categorisation up to 28%, and together they explained up to 44% of the BL variance across the Bilateria. The authors observe that these fractions of explained BL variance are much more than the one of generation time, which explained about 13% of the variability in mitogenomic evolutionary rates in non-vertebrates, and more than those of a broad range of life history and ecological variables in mammals.

The manuscript is clearly written, with an appropriate length. The results are novel, convincing, and not oversold. Sufficient methodological details are provided, allowing reproducing the analyses. I have the following major and minor comments that I hope the authors will consider in a revised, strengthened, future version of their manuscript.

MAJOR POINTS

1. All the statistical analyses heavily rely on accurate BL estimates on the phylogeny. Given the high number of sequences to be analyzed — here 10,911 — I understand that BL estimations have been conducted with FastTree 2. However, it would be important to measure how these BL estimates compare to the ones computed by standard maximum likelihood softwares (IQTREE, RAxML, PhyML, ...). If there are CPU limitations, comparisons might be conducted on smaller clades within Bilateria.
2. It is known that longer branches require more accurate models of evolution (because more unobserved changes must be inferred), so increased taxon sampling (which breaks up long branches) greatly benefits methods estimating BL (e.g., Heath et al 2008 <https://www.jse.ac.cn/fileup/1674-4918/PDF/2008-3-239-18783.pdf>). Here, the molecular evolution rate of proteins encoded by mitochondrial DNA was measured by root-to-tip distances. However, as multiple substitutions are better detected by increased taxonomic sampling, denser clades would also be those for which branch lengths are better estimated, especially those that are longer. Therefore, a denser taxonomic sampling in some clades will involve a longer trajectory from root to tip. Do the authors evaluate whether significantly elevated evolutionary rates might reflect denser availabilities of mitogenomic information in some areas of the bilaterian tree of life?
3. L184-186. At lower taxonomic levels (class and order), results were somewhat less consistent than at higher taxonomic levels, but the authors propose that analyses were weakened by low numbers of samples for many categories. This result seems rather counter-intuitive as we might expect to detect more significant biological effects at lower taxonomic scales (i.e., closer to the species level), whereas analyses at higher taxonomic scales (i.e., towards the kingdom level) might be blurred by the accumulation of events associated to greater evolutionary divergences, for instance changes in effective population size, in generation times, in life history traits, in locomotory traits, and/or in r/K strategies.

MINOR POINTS

4. I found the two abbreviations "LH" and "LC" — widely used throughout the text — quite confusing for the reader. Why not considering using something like LHT (Life History Traits) versus LOC (LOcomotory Capacities)?
5. Lines 48-51: "They found exceptionally long branches in Nematoda, Platyhelminthes, Tunicata, some Mollusca, and some Arthropoda (Acari and Copepoda). Most of these taxa comprise both free-living and parasitic lineages, but some comprise only parasitic and some only free-living species." Reformulate these two sentences to clearly state which group(s) can be tagged as parasitic / free-living only / mixed.
6. Lines 67-72. For the broad readership of Nature Communications, explain "genetic arms race", "compensation-draft feedback". Moreover, why high speciation rates are expected in parasites?
7. L73. "section" or "selection"?
8. Line 87. "fish" is a catch-all term. References 28 and 29 actually refer to teleosts.
9. Line 131. "some bats" might be clarified into "vampire bats".
10. L153. Is there any indication in Supplementary File S1:Table S2 about "statistical power"? Also see L.191-192: is there any number of species / threshold ensuring "good statistical power"?
11. L221: Figure 1. In the tree, tunicates do not appear among chordates whereas they are known to be mitochondrial fast-evolvers. Does this mean that each BL subtending each red triangle corresponds to the average BL computed over all collapsed taxa? Moreover, in the caption, it might be indicated that the source used to constrain the bilaterian phylogeny is Laumer et al. 2019.
12. L306. The "selection analyses" subsection indicates that the purifying selection pressures were relaxed in endoparasites and parasitoids — but not in ectoparasites — vs. the free-living, and in low vs. high locomotory groups. These results are not commented in the "Discussion" section. Why?
13. L537-538: "As the correlation between the two trees was very high (99.2%), we relied on the LG+G tree for downstream analyses." I assume that the correlation is not "between trees" but either between root-to-tip branches of the two trees or between each individual branch length as

estimated from the same topology under each of the two models? By far, the second option seems the best one to alleviate phylogenetic inertia.

14. L533-534. "... the LG evolutionary model, which accounts for the variability of evolutionary rates across sites in the matrix". I think that the LG model accounts for variability in amino acid exchangeabilities, not for across-site heterogeneities (this is supposed to be accommodated by the Gamma distribution).

15. Throughout the text: The word "classification" is often used, whereas one of its biological meanings is linked to "systematics" and "taxonomy". I would suggest a more frequent use of the word "categorization", especially when referring to life history and locomotory traits.

16. L533, Table S4, and comments #1 and #2: Why using JTT and LG models (for amino acid exchangeabilities) whereas some models centered on mitochondrial proteins are available? See for example the mtZOA model (Rota-Stabelli et al 2009 <https://doi.org/10.1016/j.ympev.2009.01.011>).

RESPONSE TO REVIEWERS' COMMENTS

Reviewer #1 (Remarks to the Author):

1. Overall, I found this to be an interesting and relevant study of the role of two important traits (parasitism and locomotory ability) in predicting the rate of molecular evolution in animal mitochondrial genomes. In general, I found the study to be clearly written and convincingly conducted, with a large and expansive sample size and sets of analyses (both phylogenetic and non-phylogenetic). I have a few comments that I believe will help the authors to improve their paper.

>>R: Thank you for investing your time and expertise into revising our paper, as well as for your constructive and useful critique and encouraging comments. We thoroughly and, we believe, successfully addressed all of them.

2. One of my more major comments relates to investigations and reporting upon data quality. In large, public databases, I frequently find errors, such as bacterial contaminant sequences labeled at animals. Given that such types of errors can lead to long branch lengths, and erroneous branch length estimates could have a major impact on conclusions for this study, it is important to clarify briefly (in the main manuscript) how data quality was scrutinized. For example, I suggest to look at distance matrices for each individual gene and investigate outliers (e.g. through BLAST and trees). It can also be clarified if all alignments for protein-coding genes were checked for stop codons.

>>R: This is a valid concern. We addressed it to an extent in the original study by focusing only on mitogenomes available in the curated RefSeq database. GenBank staff inspects every mitogenome belonging to a non-represented species. This includes comparing gene and protein sequence divergence, and the presence of stop codons. As a result, there should be no such major outliers (e.g. bacterial contaminant sequences) in the RefSeq database. Furthermore, we also conducted codon-based evolutionary analyses, which would immediately identify any species that had a different code (internal stop codons present in gene sequences). However, these weren't conducted on the entire data. For the revision, we conducted detailed analyses using several different approaches. Details are given in the revised Supplementary Information: Supplementary Methods: Datasets. Briefly, we used three different measures to identify outliers. We compared root-to-tip branch lengths of all species in the dataset, long-branch scores¹, and terminal branch lengths ("spurious species"). We also tested all species for the presence of stop codons. All of these analyses indicate that there are no nonbilaterian species in the dataset.

3. I suggest that it's important and interesting to discuss the potential implications of this work beyond the mitogenome. Would you expect patterns to be similar in the nuclear genome? In the discussion, I also suggest that variation in locomotory ability or dispersal distance by sex are relevant to consider, in light of the (mostly) maternal inheritance of the mitogenome.

>>R: Thank you for these two suggestions, both are relevant. In the original paper, we only briefly mentioned the fact that parasitism may affect the N_e . In the revised manuscript we elaborated on this a bit more and mentioned the role of female dispersal in it, as well as discussed the role of parasitism on other genomes. Following the reviewer's prompt, we added a brief section discussing the impact of locomotory capacity on female dispersal and mitogenomic evolution (line 367): "While parasitism may cause elevated evolutionary rates by reducing the N_e via founder effects and

increased speciation^{6,16,19}, locomotory capacity may achieve this via diminished (female) dispersal ability. However, this might depend on multiple ecological factors (e.g. the effect might be less pronounced in aquatic animals with a planktonic larva stage)". We should note here that many invertebrates are bisexual, but it is beyond the scope of this study to discuss this matter in depth. As regards the impact of parasitism on nuclear genome evolution, this is an interesting topic that remains rather poorly understood. We expect the effect of parasitism to be less strongly pronounced in nuclear genomes due to the fact that genes in the nuclear genome have a diverse range of functions, not all of which may be affected by adaptations to a parasitic lifestyle, whereas mitogenome is functionally highly specialised for energy production. This would also make it a bit more complicated to study the association of nuclear genome with life history and locomotory capacity. Although it is beyond the scope of this study, we are planning to test the same or similar hypothesis on nuclear genomes in our future studies. So far, multiple studies found that nuclear genomes tend to be reduced in size in parasitic lineages²⁻⁴. However, to our knowledge, the impact of parasitism on the nuclear evolutionary rate had not been studied systematically. We added a brief mention of the effect of parasitism on genomic evolution (line 409): "While there is evidence of a reduction in size of nuclear genome in parasitic animals³³, it remains unknown whether these increased mitogenomic evolutionary rates are also mirrored in nuclear genomes of animals. It was proposed that in parasitic plants all three genomes (nucleus, mitochondrion, and chloroplast) exhibit elevated evolutionary rates²², but this was subsequently (partially) contested⁴⁰. Therefore, multiple lines of evidence indicate that the elevation of evolutionary rates might be a common feature accompanying parasitic life history across the tree of life and different genomes, but bilaterian mitogenomes aside, the evidence remains circumstantial".

4. The conclusions would be more convincing if additional possible major covariates of molecular rate variability from the literature (e.g. body size) were included explicitly in the multivariable analyses, or, at the least, considered in turn in the discussion. Other major traits can vary across phyla. Body size has been shown to be a significant correlate of rates in birds, for example, and this trait may be especially relevant given that the majority of the data analyzed are Chordata. Benoit Nabholz, Robert Lanfear, Jérôme Fuchs. Body mass-corrected molecular rate for bird mitochondrial DNA. *Molecular Ecology*, 2016, 25 (18), pp.4438 - 4449. [ff10.1111/mec.13780](https://doi.org/10.1111/mec.13780). [ffhal-01387904f](https://doi.org/10.1111/mec.13780)

>>R: This was briefly mentioned, and the paper mentioned by the reviewer cited, in the original Supplementary file S1: "Effects of some variables appear to be lineage-specific: in vertebrates, body size/mass and metabolic rate calibrate the (mitochondrial and nuclear) molecular clock (smaller animals = faster evolution)¹⁰⁻¹², but there is no evidence for this in invertebrates^{13,14}." Therefore, it appears that the correlation between body size and mitogenomic evolutionary rate found in vertebrates breaks down in invertebrates. In a bit more detail, Thomas et al. (2006) tested a broad range of invertebrate taxa, and found no significant correlation between body size and rate of molecular evolution in any studied lineage. They concluded that "Given these data, we can confidently reject a general, genome-wide influence of body size on the rate of molecular evolution in metazoans, even though such an effect may operate locally for particular taxa, such as vertebrates, or may be evident at deeper levels of divergence" (<https://doi.org/10.1073/pnas.0510251103>). It could be because body sizes in invertebrates are difficult to compare across lineages due to diverse

range of body forms, as well as the relative scarcity of data compared to vertebrates. These findings fit our observations in flatworms, where some parasitic lineages (such as cestodes) exhibit much longer and larger bodies than free-living lineages, but parasitic flatworms exhibit more than twice longer branches than the free-living. Originally, we planned to put most of this discussion in the main document, but we felt that it made the Discussion section too long and meandering, so eventually we decided to move it to Supplementary data. Therefore, we only mention the impacts of competing variables are inconsistent and much smaller than the ones identified in our study, in the main body of the revised manuscript (line 390): “Finally, multiple variables that have been associated with mitogenomic evolution were not included in our analyses, such as generation time, longevity, body size/mass, effective population size, metabolic rate, etc. (see Supplementary Discussion for more information). However, the impacts of these competing variables are inconsistent across different lineages, especially when invertebrates are included^{13,15–17}. Also, their effect sizes were generally much smaller than the ones observed in our study whenever a broad range of lineages was included. For example, the generation time and/or longevity explained about 12-13% of the variability in mitogenomic evolutionary rates in both vertebrates¹⁸ and invertebrates¹⁹, while body size explained only 6% in mammals¹⁸ (only *cytb* was used in this study).”. For other discussions, see supplementary files.

5. typo. Intro. Line 73. “section” should apparently read “selection”.

>>R: Thanks for spotting this typo, we fixed it.

6. Methods. I would like to see more detail about the classification of the organisms into locomotory capacity groupings, as that is a tricky issue. As well, as part of full reproducibility of the work, I would like to request that the authors include the biological trait data clearly, such as a supplementary file available for download in a standard format (e.g. CSV), with the trait classifications available for all species analyzed.

>>R: A csv file with all relevant data for each species, including the life history and locomotory capacity classification, is available as Source Data: Worksheet A. Regarding the classification of locomotory capacity, we dedicated an entire section to this classification – Methods: Classification of locomotory capacity. We slightly rewrote it during the revision to improve clarity. We also mentioned this in the limitations section of the Discussion in the main manuscript (line 376): “As there are no clear lines according to which species can be categorised according to their locomotory capacity, which is a continuous trait, the Intermediate LOC category probably exhibited some overlap with the other two categories, but the introduction of this category allowed us to avoid overlap between the High and Low LOC categories. As this categorisation of locomotory capacity was insufficiently precise to capture the full range of differences in locomotory capacity between bilaterian species, it is likely that the explanatory power of this variable was underestimated in our study.” For the revised manuscript, we also added a corresponding section in Supplementary Information: Supplementary Methods: Categorisation of locomotory capacity. We provided several examples of the classification of some problematic taxa there.

7. Code. The authors mention writing custom code for these analyses. As part of transparency/reproducibility, I would recommend publishing the code as a supplementary file and/or publicly on a well-known site (e.g. on GitHub).

>>R: We added the following code availability statement: “

The code written for filtering the mitogenome data was subsequently incorporated into PhyloSuite v. 1.2.3⁴³, available from <https://github.com/dongzhang0725/PhyloSuite> or <https://pypi.org/project/PhyloSuite/>. The remaining code written for this study was deposited in Zenodo: <https://doi.org/10.5281/zenodo.7940125>.”

8. Line 154-155. What phylogenies were used to count the number of evolutionarily independent transitions to parasitism? And what method was used for the trait state mapping? Maximum parsimony mapping? Or ML or a different method?

>>R: During the revision, we conducted several rounds of additional phylogenetic analyses. After the comparison of trees, we selected a single topology to conduct all analyses and stated so clearly in the manuscript. In the Supplementary Information, we provide a detailed outline of multiple analyses that we conducted in the Supplementary Methods: Phylogenetic analyses section. It is quite long, so we decided to not paste it here. Briefly, we tested a range of methods, found that branch lengths of trees produced all exhibited correlations >90%, and selected the alignment of amino acids of 12 protein-coding genes, IQ-TREE and mtZOA+G4+F+C50 model as the optimal option. All subsequent analyses were conducted anew using this phylogeny, including the trait state mapping. This is the reason why most specific numbers are different in the revised study. Importantly, our main conclusions were not affected (they were even slightly strengthened), which indicates that our findings are robust to branch length variation across different methods. Following the reviewer’s prompt, we added a new “Supplementary Note 2. Independent origins of parasitism” to Supplementary information: “Independent origins of parasitism were inferred by inferring the ancestral states using the Maximum Likelihood method in BayesTraits²¹, and then manually inspecting the phylogenetic tree and identifying sister groups to parasitic lineages. This method is prone to phylogenetic artefacts and incomplete sampling-caused errors, so we also relied on published literature to confirm our trait state mapping. In some cases, the evolutionary scenarios were rather straightforward (e.g. a single origin in Acanthocephala), but in some cases, the evolutionary history of parasitism is very complex, so the ancestral mode is difficult to infer. Nematoda are an example. For this lineage, we inferred 7 independent origins of endoparasitism following the scenarios of the evolution of parasitism proposed for this lineage in previous studies²²⁻²⁴: in Trichinellida, Mermithida, Spirurina, Steinernematidae, Heterorhabditidae, and two in Strongylida. Another example is Platyhelminthes, where there can be 2 or 3 origins of endoparasitism, depending on the unresolved relationships between Monogenea, Cestoda and Trematoda. In Arthropoda, we identified two origins of endoparasitism (Rhinonyssidae and Pentastomida) and 12 independent origins of ectoparasitism: Lernaecidae, Psoroptidae, Ergasilidae, Caligidae, Bopyridae, Cymothoidae+Corallanidae, Hippoboscidae+Streblidae, Siphonaptera, Phthiraptera, Argulidae, Varroidae, and Ixodida.” We also provided more details in the main manuscript (line 170): “Accounting for some uncertainty in evolutionary scenarios (e.g. in Nematoda and Platyhelminthes), independent origins of endoparasitism were observed in Acanthocephala (1 origin), Arthropoda (2: Pentastomida and Rhinonyssidae), Orthonectida (1), Platyhelminthes (2 or 3),

and 5 to 7 origins in Nematoda according to the evolutionary scenario proposed previously²³. We identified 12 putative origins of ectoparasitism in Arthropoda and 1 in Platyhelminthes (see Supplementary Note 2 and Supplementary Table 3). However, low locomotory capacity was inferred as the ancestral state for Bilateria with 99.9% probability, so we inferred independent origins of high locomotory capacity. They were identified in only three phyla (Figure 1): 1) in the Cephalopoda class of Mollusca; 2) in Chordata; 3) in Arthropoda, which exhibited a complex phylogenetic distribution of different categories, with over 15 transitions between low and high locomotory capacity. “. We also cited BayesTraits in the methods section.

9. Line 160. Typo. “braches” should be “branches”

>>R: Thanks for spotting this typo, we fixed it.

10. Line 191. The authors refer to the number of species in the locomotory groups and state there is good statistical power. However, the number of evolutionarily independent transitions between locomotory groups is more relevant here than the absolute number of species and should be reported.

>>R: This is a justified objection. For the revised manuscript, we inferred Low locomotory capacity as the ancestral state with high probability (99%) using BayesTraits. This implies that we can only infer independent origins of High locomotory capacity (and Intermediate, but this is less relevant for our study, as we focused on the contrast between High and Low in most of our analyses). Aside from Arthropoda, there were only two major transitions to high locomotory capacity: in Chordata and in Cephalopoda. Arthropoda exhibited a complex evolutionary scenario, with over 20 transitions between low and high locomotory capacity. We also provided pie charts in the revised Figure 1 where readers can infer the distribution of locomotory capacity and life history categories in each phylum. As mentioned above, we added the following to the text (line 187): “Low locomotory capacity was inferred as the ancestral state for Bilateria with 99.9% probability, so we inferred independent origins of high locomotory capacity. They were identified in only three phyla (Figure 1): 1) in the Cephalopoda class of Mollusca; 2) in a majority of Chordata; 3) in a majority of Arthropoda, which exhibited a complex phylogenetic distribution of different categories, requiring at least 15 High ↔ Low transitions.”.

11. I would find Figure 1 to be more informative with regards to this study if the tips were colour coded in a different way... perhaps the tips could be pies or bars with the proportion of species studied in each phylum with each trait. Or, the tree could be presented side by side with the trait information. (Such a plot can be made in various ways, but one option to consider is to create an informative phylogeny + trait figure using the R package ggtree.)

>>R: Thank you for this suggestion. We revised the figure during the revision according to your comment. Furthermore, we merged all of the data in the original Table 1 into the revised Figure 1, which allowed us to trim the manuscript a bit. Data from the original Table 1 were moved to Supplementary data and merged with some other data (now Supplementary Table 3).

12. The version number and date of access should be given when citing an online tutorial.

>>R: We added the version and date of last access to the revised manuscript: “(version: 7/7/2016; last accessed 30/3/2023)”.

13. Clarify the definition of used to identify statistical outliers. Also, were single-gene trees investigated for outliers in addition to the overall tree?

>>R: In the revised MS, we defined branch length outliers in the dataset as “1.5 times the interquartile range” and stated that they were identified using the “boxplot” function in R. Single-gene trees were not investigated for branch length outliers as this was not crucial for our study, which was primarily interested in the evolution of complete mitogenomes. Also, the identification of outliers in the overall tree (concatenated PCGs) revealed that all endoparasites were outliers. This confirms that the outliers are not caused by sequencing or annotation errors. Instead they are a normal part of the natural data distribution in this case. We should also note that the negative impact of outliers is inversely correlated with the sample size, and our sample was over 10,000, which indicates that analyses should be rather robust. We did confirm that single-gene topologies produce congruent results, though. Branch length correlations were rather high, apart from three rapidly-evolving genes. Even for these, we found that our main conclusions still hold. These analyses can be found in Supplementary Information: Supplementary Note 6. Single gene results.

14. The phylogenetic level at which the substantial explanatory power was found needs to be mentioned in the abstract. Some other traits have been shown to have very strong explanatory power within some groups. Therefore, it is important to clarify up front the nature of the findings and the level at which they apply. I suggest it is important not to accidentally imply that rates don't vary a lot within some groups in association with traits.

>>R: We rewrote the manuscript in multiple places to improve the clarity in this aspect. We clearly discuss the four outlier lineages, both in Results and Discussion. We also made it clear in the abstract that we refer to the overall bilaterian dataset (due to the word count limitations imposed by the journal, there was no space to mention the outliers in the Abstract).

15. I would find it easier to draw conclusions from Table 2 if the results were presented in figure format (with the full information kept as either table or supplementary file). For example, perhaps the taxa could be along the x-axis, and the y-axis could show the p-values or R² values. The different variables (and combination of variables) could be shown as symbol/colour combinations. It would be easier to draw a conclusion about the relative explanatory strength of the variables, as well as the consistency of the results across taxa, if the information were to be presented in visual format.

>>R: In the revised manuscript, we followed the reviewer's suggestion and presented the data in figure format (Figure 3).

Again, I enjoyed reading this paper, and I hope that the authors and editor will consider my comments for improving the research further.

R: Thank you once again for your encouraging comments and constructive critique. They helped us to improve our paper, and we did our best to fully address all of them.

Reviewer #2 (Remarks to the Author):

In this interesting manuscript, Ivan Jakovlić and colleagues study the determinants of the evolution of the proteins encoded by the mitochondrial genome of Bilaterians. The underlying logic of the authors is that the mitogenome plays a central role in energy production thanks to its coding capacities for 13 proteins involved in the respiratory chain (CYB, COXI-II-III, NAD1-2-3-4-4L-5-6, and ATP6-8). Consequently, the strength of purifying selection acting at the DNA level on mitochondrial genes should be positively correlated with the energy expenditure of the organisms. The latter would in turn produce a positive correlation with the strength of selection for locomotory capacity. To test these hypotheses, the authors assemble a dataset of life history traits (parasitic versus free-living), locomotory capacity traits, and mitochondrial proteins for 10,911 bilaterian taxa. From the mitochondrial alignments, they estimate branch lengths (BL) on a reference phylogenetic tree. Their statistical analyses evaluate the degree of correlation between these BL and the life history and locomotory capacity traits. Their main finding is that the proportion of BL variance explained by the locomotory traits is on average (much) smaller than the proportion of BL variance explained by the life history traits. Therefore, the locomotory capacity is not the only explanatory variable for elevated evolutionary rates in parasitic lineages. Of note, parasitic taxa possessing a free-living (adult) stage are evolving under selection for higher locomotory capacity. Importantly, life history categorisation (parasitism) explained up to 37%, locomotory categorisation up to 28%, and together they explained up to 44% of the BL variance across the Bilateria. The authors observe that these fractions of explained BL variance are much more than the one of generation time, which explained about 13% of the variability in mitogenomic evolutionary rates in non-vertebrates, and more than those of a broad range of life history and ecological variables in mammals.

The manuscript is clearly written, with an appropriate length. The results are novel, convincing, and not oversold. Sufficient methodological details are provided, allowing reproducing the analyses. I have the following major and minor comments that I hope the authors will consider in a revised, strengthened, future version of their manuscript.

>>R: Thank you for investing your time and expertise into revising our paper, as well as for your constructive and useful critique and encouraging comments. We believe that we have successfully addressed all of them, and we provide detailed explanations of how we did that below.

MAJOR POINTS

1. All the statistical analyses heavily rely on accurate BL estimates on the phylogeny. Given the high number of sequences to be analyzed — here 10,911 — I understand that BL estimations have been conducted with FastTree 2. However, it would be important to measure how these BL estimates compare to the ones computed by standard maximum likelihood softwares (IQTREE, RAxML,

PhyML, ...). If there are CPU limitations, comparisons might be conducted on smaller clades within Bilateria.

>>R: Following this suggestion, we conducted several rounds of Maximum Likelihood algorithm-based phylogenetic analyses using IQ-TREE during the revision. We present all steps in detail in the Supplementary Information: Supplementary methods: Phylogenetic analyses section. It is quite long, so we decided not to paste it here. Briefly, we tested FastTree and IQ-TREE programs, in combination with different datasets, and several different models. For IQ-TREE (in our experience this program produces more reliable results, but it is somewhat slower), we tested three different datasets, two modes of operation (fast and standard), and two different evolutionary models. Correlations between root-to-tip branch lengths among all trees were between 90% and 99%, which indicates that any methodology would produce congruent overall results. After careful comparison of topologies, we selected the alignment of amino acids of 12 protein-coding genes, IQ-TREE and mtZOA+G4+F+C50 model as the optimal option. We selected the C50 evolutionary model because it is a variant of the CAT model, designed to account for substitutional heterogeneity²⁵, and it is expected to produce more accurate branch length estimates in deep phylogenies due to improved modelling of mutational convergences and reversions²⁶. This may be particularly important for our dataset, comprising very deep evolutionary splits. We reconducted all analyses using this tree, so most specific numbers are different in the revised manuscript. Despite this, none of our major conclusions were affected. This further confirms that our findings are robust regardless of the algorithm used to infer the branch lengths. Notably, these new, arguably better, branch lengths, further increased the explanatory power of LHT and LOC by a few points. The life history categorisation (parasitism) now explains ≈45%, locomotory capacity categorisation ≈39%, and together they explain ≈56% of the total variability in evolutionary rates of mitochondrial protein-coding genes in Bilateria. Also, the new selection pressure analyses are in better agreement with branch lengths, which may indicate that new, improved, branch lengths improved the performance of HyPhy algorithms.

2. It is known that longer branches require more accurate models of evolution (because more unobserved changes must be inferred), so increased taxon sampling (which breaks up long branches) greatly benefits methods estimating BL (e.g., Heath et al 2008 <https://www.jse.ac.cn/fileup/1674-4918/PDF/2008-3-239-18783.pdf>). Here, the molecular evolution rate of proteins encoded by mitochondrial DNA was measured by root-to-tip distances. However, as multiple substitutions are better detected by increased taxonomic sampling, denser clades would also be those for which branch lengths are better estimated, especially those that are longer. Therefore, a denser taxonomic sampling in some clades will involve a longer trajectory from root to tip. Do the authors evaluate whether significantly elevated evolutionary rates might reflect denser availabilities of mitogenomic information in some areas of the bilaterian tree of life?

>>R: This is a valid concern. However, it is highly unlikely that this is the case, as a vast majority of species belonged to Chordata and Arthropoda, most of which are free-living, and both of which exhibited some of the shortest branches in the dataset. On this basis, we expect a negative correlation between branch lengths and taxonomic sampling density. To test this hypothesis, we inferred the correlation between the number of species per phylum and the average root-to-tip branch length per phylum. We added a brief discussion of this problem to the Supplementary

Discussion: “As reverse substitutions are better detected by increased taxonomic sampling, denser-sampled clades might also exhibit longer branches. To test whether this may have affected our results, we inferred the correlation between the number of species per phylum and the average root-to-tip branch length per phylum. The correlation coefficient was negative ($r = -0.23$) and nonsignificant (p -value = 0.27). Indeed, a vast majority of species belonged to Chordata and Arthropoda, most of which are free-living, and both of which exhibited some of the shortest branches in the dataset. Therefore, we can reject the hypothesis that our findings are an artefact caused by a better sampling of parasitic species. “

3. L184-186. At lower taxonomic levels (class and order), results were somewhat less consistent than at higher taxonomic levels, but the authors propose that analyses were weakened by low numbers of samples for many categories. This result seems rather counter-intuitive as we might expect to detect more significant biological effects at lower taxonomic scales (i.e., closer to the species level), whereas analyses at higher taxonomic scales (i.e., towards the kingdom level) might be blurred by the accumulation of events associated to greater evolutionary divergences, for instance changes in effective population size, in generation times, in life history traits, in locomotory traits, and/or in r/K strategies.

>>R: As there are many analyses at lower levels, we did not want to mention minor outliers in the main manuscript in detail. By minor outliers, we refer to those different from the overall bilaterian patterns, but not affecting our working hypotheses. For example, the relationship of microparasites and free-living species was complicated at lower taxonomic levels, with all three combinations observed: microparasites exhibiting longer branches, no statistical difference between the two, and shorter branches in microparasites. However, branch lengths of these two categories did not differ at the level of Bilateria, and this is not central to our working hypothesis, which merely claims that nominally parasitic life history itself does not provide a convincing reason for elevated evolutionary rates in these lineages. As the reviewer mentioned, there could be a plethora of other reasons, such as fluctuations in N_e , causing these differences. While the reviewer’s argument makes sense, there is one factor that may also make such small lineages more prone to random fluctuations: the lower the phylogenetic level, the more likely that there will be a single origin of the trait of interest (parasitism or high/low locomotory capacity in this case). This relatively close shared ancestry of studied lineages can also easily disrupt the evolutionary patterns observed at higher levels. Furthermore, and even more importantly, as the numbers of species included in at least one category compared were commonly rather small at lower taxonomic levels, these may easily be merely statistical artefacts caused by sparse sampling. Very small samples can easily be nonrepresentative of the overall population, which then vastly increases the chances of random statistical fluctuations producing artefactual results. For example, in many cases, we had less than ten species in a lineage. Compared to lineages where we had triple-digit numbers of species, we put far less faith in the single-digit lineages. Another factor that hampered these analyses was the fact that many categories were not included at lower taxonomic levels. Importantly, our central hypotheses were supported in the majority of cases: branches were consistently longer in endoparasites and ectoparasites than in free-living, and longer in the Low locomotory capacity category than in High locomotory capacity. The only exceptions in life history categorisation were Nematoda and Arachnida, which we discuss at length and explain that selection for locomotory capacity may help us explain these outliers. There were a few minor exceptions at lower taxonomic

levels, where only 1 to 3 species were included in the dataset. In the LOC categorisation, the outlier was Hexanauplia, and Diptera was an exception in the LHT categorisation. The small number of samples in key categories prevents us from making any conclusions about these two lineages, i.e. we cannot assess whether they are true outliers or merely statistical flukes caused by small samples and weak statistical power. We rewrote the text in a way to clearly highlight the outliers and provided additional details in the supplementary file.

MINOR POINTS

4. I found the two abbreviations "LH" and "LC" — widely used throughout the text — quite confusing for the reader. Why not considering using something like LHT (Life History Traits) versus LOC (LOcomotory Capacities)?

>>R: We revised these abbreviations according to the reviewer's suggestion throughout the materials.

5. Lines 48-51: "They found exceptionally long branches in Nematoda, Platyhelminthes, Tunicata, some Mollusca, and some Arthropoda (Acari and Copepoda). Most of these taxa comprise both free-living and parasitic lineages, but some comprise only parasitic and some only free-living species." Reformulate these two sentences to clearly state which group(s) can be tagged as parasitic / free-living only / mixed.

>>R: We revised the sentence as advised (line 46): "In a meta-analysis of all bilaterian mitogenomes available in 2012, Bernt et al. ¹² found exceptionally long branches in several lineages, some of which comprised both free-living and parasitic species (Nematoda, Platyhelminthes, Acari, and Copepoda), but other comprised only free-living (Tunicata and Mollusca)."

6. Lines 67-72. For the broad readership of Nature Communications, explain "genetic arms race", "compensation-draft feedback". Moreover, why high speciation rates are expected in parasites?

>>R: Here we are presenting explanations proposed by other scientists, which does not necessarily mean that we agree with them; and in the discussion we briefly mentioned what we consider to be weaknesses of these explanations. In this particular case, Castro et al. ²⁷ proposed that: "On the basis of our results, the most likely explanation for the increased rate of molecular evolution in the Hymenoptera is their increased rate of speciation compared with the dipteran parasitic lineages." It has been proposed that rapid host-parasite arms race-driven genetic adaptation may lead to increased speciation rates in parasites and that parasites exhibit an increased potential for sympatric speciation ²⁸. Some scientists even made rather bold claims about parasite diversity outmatching that of free-living species ²⁹, and others appear to agree that most extraordinary adaptive radiations have been observed in parasitic organisms ²⁸. However, some studies failed to find evidence for the hypothesis of increased rates of speciation in parasites ³⁰ and the hypothesis that cryptic taxa are more frequent among parasitic than free-living taxa ³¹. As this was not a crucial issue for our study,

we did not debate it at length. We rewrote this section by adding brief explanations for the arms race and compensation-draft, and indicating that increased speciation rates remain somewhat speculative (line 64): “Previously proposed candidates include directional selection driven by the genetic arms race between hosts and parasites (adaptations and counter-adaptations in host-parasite co-evolution)^{32,33}, the compensation-draft feedback (fixation of mildly deleterious mutations results in selection for compensatory mutations, which lead to fixation of additional deleterious mutations in nonrecombining mitochondrial genomes)³³, or the reduction of purifying selection pressures in parasitic species, which may be driven by the effective population size reductions³⁴ putatively driven by high speciation rates in parasites and/or frequent founder events during transmissions to new host individuals^{27,32,35}. “

7. L73. "section" or "selection"?

>>R: Thank you for spotting this typo. It was supposed to be selection. We fixed it.

8. Line 87. "fish" is a catch-all term. References 28 and 29 actually refer to teleosts.

>>R: Thanks, we added “teleost” here.

9. Line 131. "some bats" might be clarified into "vampire bats".

>>R: Changed as suggested.

10. L153. Is there any indication in Supplementary File S1:Table S2 about "statistical power"? Also see L.191-192: is there any number of species / threshold ensuring "good statistical power"?

>>R: For the revised manuscript, we conducted statistical power analyses for the main two datasets: Bilateria, LHT and LOC categorisation; as well as for the four outlier taxa. We added the results to Supplementary Figure 1 (the one containing the effect size analyses). As power depends on the sample size and the effect size, the power was very good (1.0) in most major pairwise comparisons, apart from the few where the effect size was small. Luckily, these were also less important analyses for our study, such as the comparison between free-living and micropredatory lineages. In cases where we did not conduct power analyses, we changed the wording in the text (did not use the word “power”).

11. L221: Figure 1. In the tree, tunicates do not appear among chordates whereas they are known to be mitochondrial fast-evolvers. Does this mean that each BL subtending each red triangle corresponds to the average BL computed over all collapsed taxa? Moreover, in the caption, it might be indicated that the source used to constrain the bilaterian phylogeny is Laumer et al. 2019.

>>R: Indeed, the topology shows average root-to-tip branch lengths per phylum. As Tunicata is a subphylum, we included it in chordates. 35 Tunicata species (all Ascidiacea) were included in the dataset. All were classified as free-living and all as Low LOC. Therefore, they indirectly support our

hypothesis that low LOC allows relaxed purifying selection pressures and increased evolutionary rates. During the revision, we added the reference (Laumer et al.) and clarified other issues. Also, this figure was reworked during the revision; we added the data initially included in Table 1, so the caption underwent a major overhaul as well.

12. L306. The "selection analyses" subsection indicates that the purifying selection pressures were relaxed in endoparasites and parasitoids — but not in ectoparasites — vs. the free-living, and in low vs. high locomotory groups. These results are not commented in the "Discussion" section. Why?
>>R: As we did not have full confidence in these results (we outline the reasons below and in several places in the manuscript), to avoid cluttering the manuscript with non-crucial details, we opted to mention these results only fleetingly. In the revised manuscript, we added a longer description of results, and mentioned all of them at least briefly. However, we still opted to keep the table and more detailed description of results in supplementary data, as these results were inferred on a fragment of the overall dataset due to the computational and algorithm limitations of HYPHY, and we are afraid that highly divergent genes make it difficult to infer homology with confidence, so we do not have full confidence in the reliability of these analyses. Especially as selection analyses are known to produce unreliable results across different algorithms³⁶. In agreement with this, we observed that less phylogenetically diverse datasets (spanning a single phylum) produced results that were in better agreement with branch length patterns than datasets that comprised species belonging to a wide range of different phyla (please see Supplementary Table 8). We can conclude with some confidence that such tests are probably more reliable on closely phylogenetically related datasets. Also, we used new branch lengths to re-conduct these analyses, which affected some of the results, including some of those highlighted by the reviewer: selection pressures are now relaxed in ectoparasites vs. free-living, but not in Low LOC vs High LOC groups when representatives of all relevant bilaterian lineages were included; when only representatives of Arthropoda were included, Low LOC group exhibited slightly but significantly relaxed purifying selection pressures. Again, due to the reasons outlined above, and the sensitivity of results to multiple variables, we decided to limit our discussion of these results and put them in supplementary data.

13. L537-538: "As the correlation between the two trees was very high (99.2%), we relied on the LG+G tree for downstream analyses." I assume that the correlation is not "between trees" but either between root-to-tip branches of the two trees or between each individual branch length as estimated from the same topology under each of the two models? By far, the second option seems the best one to alleviate phylogenetic inertia.

>>R: The reviewer is correct in pointing out that our shortcut lacked precision. Indeed, the correlation refers to the root-to-tip branch length of each individual leaf compared between two trees inferred using different methodologies. We edited the text accordingly. Also, during the revision, we added several other phylogenetic trees, so the text was rewritten accordingly. We described this elsewhere in this response letter.

14. L533-534. "... the LG evolutionary model, which accounts for the variability of evolutionary rates across sites in the matrix". I think that the LG model accounts for variability in amino acid

exchangeabilities, not for across-site heterogeneities (this is supposed to be accommodated by the Gamma distribution).

>>R: Indeed, we made a mistake here. Thank you for pointing it out. We fixed it. For the revised manuscript, as already elaborated in response to question 1, we conducted a number of different phylogenetic analyses using different models and strategies, so we merely provided an overview in the main text, and put details in Supplementary Information (Supplementary methods: Phylogenetic analysis).

15. Throughout the text: The word "classification" is often used, whereas one of its biological meanings is linked to "systematics" and "taxonomy". I would suggest a more frequent use of the word "categorization", especially when referring to life history and locomotory traits.

>>R: Revised as advised throughout the manuscript.

16. L533, Table S4, and comments #1 and #2: Why using JTT and LG models (for amino acid exchangeabilities) whereas some models centered on mitochondrial proteins are available? See for example the mtZOA model (Rota-Stabelli et al 2009 <https://doi.org/10.1016/j.ympcv.2009.01.011>).

>>R: For the revised manuscript, we added an analysis conducted using the Maximum Likelihood approach implemented in IQ-TREE with the mtZOA model. We also added an analysis using mtZOA in combination with a "heterogeneous" C50 model (described in detail in response to question 1). All topologies exhibited highly correlated branch lengths, and our conclusions were not affected.

References

1. Struck, T. H. TreSpEx—Detection of Misleading Signal in Phylogenetic Reconstructions Based on Tree Information. *Evol Bioinform Online* **10**, 51–67 (2014).
2. Poulin, R. & Randhawa, H. S. Evolution of parasitism along convergent lines: from ecology to genomics. *Parasitology* **142**, S6–S15 (2015).
3. Sun, G. *et al.* Large-scale gene losses underlie the genome evolution of parasitic plant *Cuscuta australis*. *Nat Commun* **9**, 2683 (2018).
4. Adams, P. E., Bubrig, L. T. & Fierst, J. L. Genome Evolution: On the Road to Parasitism. *Current Biology* **30**, R272–R274 (2020).
5. Jennings, J. B. Nutritional and respiratory pathways to parasitism exemplified in the Turbellaria. *International Journal for Parasitology* **27**, 679–691 (1997).
6. Keeling, P. J. *et al.* The Reduced Genome of the Parasitic Microsporidian *Enterocytozoon bieneusi* Lacks Genes for Core Carbon Metabolism. *Genome Biol Evol* **2**, 304–309 (2010).
7. Bromham, L., Cowman, P. F. & Lanfear, R. Parasitic plants have increased rates of molecular evolution across all three genomes. *BMC Evol Biol* **13**, 1–11 (2013).
8. Haraguchi, Y. & Sasaki, A. Host–Parasite Arms Race in Mutation Modifications: Indefinite Escalation Despite a Heavy Load? *Journal of Theoretical Biology* **183**, 121–137 (1996).

9. Dawkins, R. & Krebs, J. R. Arms races between and within species. *Proceedings of the Royal Society of London. Series B, Containing papers of a Biological character. Royal Society (Great Britain)* **205**, 489–511 (1979).
10. Martin, A. P. & Palumbi, S. R. Body size, metabolic rate, generation time, and the molecular clock. *PNAS* **90**, 4087–4091 (1993).
11. Gillooly, J. F., Allen, A. P., West, G. B. & Brown, J. H. The rate of DNA evolution: Effects of body size and temperature on the molecular clock. *PNAS* **102**, 140–145 (2005).
12. Nabholz, B., Lanfear, R. & Fuchs, J. Body mass-corrected molecular rate for bird mitochondrial DNA. *Molecular Ecology* **25**, 4438–4449 (2016).
13. Thomas, J. A., Welch, J. J., Woolfit, M. & Bromham, L. There is no universal molecular clock for invertebrates, but rate variation does not scale with body size. *PNAS* **103**, 7366–7371 (2006).
14. Lanfear, R., Thomas, J. A., Welch, J. J., Brey, T. & Bromham, L. Metabolic rate does not calibrate the molecular clock. *PNAS* **104**, 15388–15393 (2007).
15. Bazin, E., Glémin, S. & Galtier, N. Population Size Does Not Influence Mitochondrial Genetic Diversity in Animals. *Science* **312**, 570–572 (2006).
16. Jakovlić, I. *et al.* Slow crabs - fast genomes: locomotory capacity predicts skew magnitude in crustacean mitogenomes. *Molecular Ecology* **30**, 5488–5502 (2021).
17. Nabholz, B., Glémin, S. & Galtier, N. The erratic mitochondrial clock: variations of mutation rate, not population size, affect mtDNA diversity across birds and mammals. *BMC Evol Biol* **9**, 54 (2009).
18. Nabholz, B., Mauffrey, J.-F., Bazin, E., Galtier, N. & Glemin, S. Determination of Mitochondrial Genetic Diversity in Mammals. *Genetics* **178**, 351–361 (2008).
19. Thomas, J. A., Welch, J. J., Lanfear, R. & Bromham, L. A Generation Time Effect on the Rate of Molecular Evolution in Invertebrates. *Mol Biol Evol* **27**, 1173–1180 (2010).
20. Xiang, C. *et al.* Using PhyloSuite for molecular phylogeny and tree-based analyses. *iMeta* e87 (2023) doi:10.1002/imt2.87.
21. Meade, A. & Pagel, M. Ancestral State Reconstruction Using BayesTraits. in *Environmental Microbial Evolution: Methods and Protocols* (ed. Luo, H.) 255–266 (Springer US, 2022). doi:10.1007/978-1-0716-2691-7_12.
22. Blaxter, M. L. *et al.* A molecular evolutionary framework for the phylum Nematoda. *Nature* **392**, 71–75 (1998).
23. Blaxter, M. & Koutsovoulos, G. The evolution of parasitism in Nematoda. *Parasitology* **142**, S26–S39 (2015).
24. Weinstein, S. B. & Kuris, A. M. Independent origins of parasitism in Animalia. *Biology Letters* **12**, 20160324 (2016).
25. Lartillot, N. & Philippe, H. A Bayesian Mixture Model for Across-Site Heterogeneities in the Amino-Acid Replacement Process. *Molecular Biology and Evolution* **21**, 1095–1109 (2004).
26. Lartillot, N., Brinkmann, H. & Philippe, H. Suppression of long-branch attraction artefacts in the animal phylogeny using a site-heterogeneous model. *BMC Evolutionary Biology* **7**, S4 (2007).
27. Castro, L. R., Austin, A. D. & Dowton, M. Contrasting Rates of Mitochondrial Molecular Evolution in Parasitic Diptera and Hymenoptera. *Mol Biol Evol* **19**, 1100–1113 (2002).
28. Huyse, T., Poulin, R. & Théron, A. Speciation in parasites: a population genetics approach. *Trends in Parasitology* **21**, 469–475 (2005).
29. Windsor, D. A. Controversies in parasitology, Most of the species on Earth are parasites. *International Journal for Parasitology* **28**, 1939–1941 (1998).

30. Medina, I. & Langmore, N. E. Coevolution is linked with phenotypic diversification but not speciation in avian brood parasites. *Proc. R. Soc. B.* **282**, 20152056 (2015).
31. Poulin, R. & Pérez-Ponce de León, G. Global analysis reveals that cryptic diversity is linked with habitat but not mode of life. *Journal of Evolutionary Biology* **30**, 641–649 (2017).
32. Downton, M. & Austin, A. D. Increased genetic diversity in mitochondrial genes is correlated with the evolution of parasitism in the Hymenoptera. *J Mol Evol* **41**, 958–965 (1995).
33. Oliveira, D. C. S. G., Raychoudhury, R., Lavrov, D. V. & Werren, J. H. Rapidly Evolving Mitochondrial Genome and Directional Selection in Mitochondrial Genes in the Parasitic Wasp *Nasonia* (Hymenoptera: Pteromalidae). *Mol Biol Evol* **25**, 2167–2180 (2008).
34. Ohta, T. Population size and rate of evolution. *J Mol Evol* **1**, 305–314 (1972).
35. Page, R. D. M., Lee, P. L. M., Becher, S. A., Griffiths, R. & Clayton, D. H. A Different Tempo of Mitochondrial DNA Evolution in Birds and Their Parasitic Lice. *Molecular Phylogenetics and Evolution* **9**, 276–293 (1998).
36. Chen, Q. *et al.* Two decades of suspect evidence for adaptive molecular evolution—negative selection confounding positive-selection signals. *National Science Review* **9**, nwab217 (2022).

REVIEWERS' COMMENTS

Reviewer #3 (Remarks to the Author):

In this revised manuscript, Jakovlic have made considerable effort to address the comments of the initial pair of reviewers. Having read the revised manuscript and the authors' response, my opinion is that the concerns of the reviewers have been well addressed. The study provides a very interesting investigation of the key factors that drive variation in mitogenomic rates in bilaterian animals, and makes some important progress in resolving some of the conflicts among previous studies.

I have a few relatively minor comments on the revised manuscript for the authors to consider.

Abstract

The first sentence of the Abstract mentions that previous evidence of elevated evolutionary rates is limited to Arthropoda, but this overlooks the findings of elevated rates in parasitic plants (which the authors note at the end of the main text).

Introduction

Line 42. The Introduction quickly moves into the specifics of previous studies, but I feel that it would be more useful to readers if the first parts of the Introduction were restructured. It would be better to start with more details about parasitism and why we might expect an impact on evolutionary rates (this is mentioned in the second paragraph but would be more effective if placed earlier).

Line 69. In this section, the phrasing implies that purifying selection is reduced because of reductions in effective population size. This is slightly misleading here for two reasons. First, reductions in effective population size leads to an increasing dominance of genetic drift (rather than specifically reducing purifying selection, although this is later clarified on line 112). Second, relaxed purifying selection is conceivably more importantly associated with other aspects of parasitism rather than effective population size (as clarified later on line 72).

Results

Line 236. The purpose of the single-gene analyses is not immediately clear to me. These would only be useful if the authors were interested in identifying specific genes that have a greater association with locomotory capacity, parasitism, etc.

Discussion

Line 411. The comparison with parasitic plants is potentially useful because the elevated rates in parasitic plants are believed to be due to relaxed selection (particularly in holoparasites), and because there are no locomotory differences among plants (although there are obviously other biological factors that are specific to plants). It would be helpful to provide greater discussion of the results from plants, for example when interpreting the evidence of relaxed purifying selection (line 364).

Figure 1. The clade labels "Ecdysozoa" and "Lophotrochozoa" seem unnecessarily and distractingly large.

RESPONSE TO REVIEWERS' COMMENTS

Reviewer #3 (Remarks to the Author):

In this revised manuscript, Jakovlic have made considerable effort to address the comments of the initial pair of reviewers. Having read the revised manuscript and the authors' response, my opinion is that the concerns of the reviewers have been well addressed. The study provides a very interesting investigation of the key factors that drive variation in mitogenomic rates in bilaterian animals, and makes some important progress in resolving some of the conflicts among previous studies.

I have a few relatively minor comments on the revised manuscript for the authors to consider.

>>R: Thank you for your comments, and for investing your time and expertise into reviewing our manuscript. We addressed your comments, which comprised slightly reorganizing and rewriting the Introduction and Discussion. None of the results and conclusions were affected.

Abstract

The first sentence of the Abstract mentions that previous evidence of elevated evolutionary rates is limited to Arthropoda, but this overlooks the findings of elevated rates in parasitic plants (which the authors note at the end of the main text).

>>R: Plants are not included in parasites as defined for the purpose of our study, so we decided to not mention this in the abstract. Instead, we changed the sentence in the following way: "The evidence that parasitic animals exhibit elevated evolutionary rates is inconsistent and limited to Arthropoda."

Introduction

Line 42. The Introduction quickly moves into the specifics of previous studies, but I feel that it would be more useful to readers if the first parts of the Introduction were restructured. It would be better to start with more details about parasitism and why we might expect an impact on evolutionary rates (this is mentioned in the second paragraph but would be more effective if placed earlier).

>>R: We rearranged the Introduction accordingly. As the logical structure changed, some sentences had to be rewritten. All changes in the manuscript have been tracked.

Line 69. In this section, the phrasing implies that purifying selection is reduced because of reductions in effective population size. This is slightly misleading here for two reasons. First, reductions in effective population size leads to an increasing dominance of genetic drift (rather than specifically reducing purifying selection, although this is later clarified on line 112). Second, relaxed purifying selection is conceivably more importantly associated with other aspects of parasitism rather than effective population size (as clarified later on line 72).

>>R: Here we want to highlight that previous studies offered only reductions in the effective population size as the explanation for relaxed purifying selection pressures. A major novelty of our study is that we offer reduced locomotory capacity as a major factor contributing to this relaxation. Also, reduction of metabolic functions was proposed as an explanatory variable in plants, but previous relevant animal studies appear to have overlooked it. We rewrote the Introduction in a way to address the reviewer's comment and clarify these issues. The sentence pointed out by the reviewer as problematic was reframed

in the following way: “Previously proposed explanations include (1) directional selection driven by the genetic arms race between hosts and parasites, involving adaptations and counter-adaptations in host-parasite co-evolution ^{6,11,12}, (2) the compensation-draft feedback, where the fixation of mildly deleterious mutations results in the selection for compensatory mutations, which lead to the fixation of additional deleterious mutations in nonrecombining mitochondrial genomes ¹¹, and (3) increased drift associated with reductions in the effective population size (N_e) ¹³, putatively caused by high speciation rates in parasites and/or frequent founder events during transmissions to new host individuals ^{6,14,15}.”

Results

Line 236. The purpose of the single-gene analyses is not immediately clear to me. These would only be useful if the authors were interested in identifying specific genes that have a greater association with locomotory capacity, parasitism, etc.

>>R: We mostly agree with the reviewer, but the main purpose of this analysis was to preemptively address a putative criticism by reviewers and readers, and show that our findings hold even when we break down the dataset into individual genes. In other words, single-gene results prove that the overall pattern of mitogenomic evolution is not produced by a single or a few genes. Instead, practically all genes produce the same pattern. Therefore, after a brief discussion among the authors, we decided to keep these data in the manuscript. They are only mentioned in a single sentence in the main body, so they don't take up much space.

Discussion

Line 411. The comparison with parasitic plants is potentially useful because the elevated rates in parasitic plants are believed to be due to relaxed selection (particularly in holoparasites), and because there are no locomotory differences among plants (although there are obviously other biological factors that are specific to plants). It would be helpful to provide greater discussion of the results from plants, for example when interpreting the evidence of relaxed purifying selection (line 364).

>>R: We reorganized the Discussion slightly following this comment. We moved the discussion of evolution in plant parasites from the last paragraph to the one indicated by the reviewer and further expanded it. As the final paragraph felt clumsy after the change, we moved another sentence to the same section of discussion, and completely discarded the final paragraph: “In partial agreement with these findings, although plants do not exhibit variability in locomotory capacity across different life history strategies, elevated evolutionary rates have been proposed for all three genomes (nucleus, mitochondrion, and chloroplast) in parasitic plants³¹. The association between the metabolic dependence on the host and the strength of purifying selection was invoked as an explanation for this observation ³¹, but a follow-up study failed to find evidence for elevated evolutionary rates in plant mitogenomes ⁴⁰. Notably, there is also evidence of a reduction in size of nuclear genome in parasitic animals ³³, but it remains unknown whether these increased mitogenomic evolutionary rates are also mirrored in nuclear genomes of animals.”

Figure 1. The clade labels “Ecdysozoa” and “Lophotrochozoa” seem unnecessarily and distractingly large.

>>R: We revised the figure accordingly.